# The Impact of YRNAs on HNSCC and HPV Infection

**DOI:** 10.3390/biomedicines11030681

**Published:** 2023-02-23

**Authors:** Kacper Guglas, Tomasz Kolenda, Joanna Kozłowska-Masłoń, Patricia Severino, Anna Teresiak, Renata Bliźniak, Katarzyna Lamperska

**Affiliations:** 1Laboratory of Cancer Genetics, Greater Poland Cancer Centre, Garbary Street 15, 61-866 Poznan, Poland; 2Postgraduate School of Molecular Medicine, Medical University of Warsaw, Zwirki and Wigury Street 61, 02-091 Warsaw, Poland; 3Institute of Human Biology and Evolution, Faculty of Biology, Adam Mickiewicz University, Uniwersytetu Poznańskiego 6, 61-614 Poznan, Poland; 4Albert Einstein Research and Education Institute, Hospital Israelita Albert Einstein, Sao Paulo 05652-900, Brazil

**Keywords:** HPV, YRNA, HNSCC, short non-coding RNA, deconvolution, NGS, biomarker

## Abstract

HPV infection is one of the most important risk factors for head and neck squamous cell carcinoma among younger patients. YRNAs are short non-coding RNAs involved in DNA replication. YRNAs have been found to be dysregulated in many cancers, including head and neck squamous cell carcinoma (HNSCC). In this study, we investigated the role of YRNAs in HPV-positive HNSCC using publicly available gene expression datasets from HNSCC tissue, where expression patterns of YRNAs in HPV(+) and HPV(−) HNSCC samples significantly differed. Additionally, HNSCC cell lines were treated with YRNA1-overexpressing plasmid and RNA derived from these cell lines was used to perform a NGS analysis. Additionally, a deconvolution analysis was performed to determine YRNA1’s impact on immune cells. YRNA expression levels varied according to cancer pathological and clinical stages, and correlated with more aggressive subtypes. YRNAs were mostly associated with more advanced cancer stages in the HPV(+) group, and YRNA3 and YRNA1 expression levels were found to be correlated with more advanced clinical stages despite HPV infection status, showing that they may function as potential biomarkers of more advanced stages of the disease. YRNA5 was associated with less-advanced cancer stages in the HPV(−) group. Overall survival and progression-free survival analyses showed opposite results between the HPV groups. The expression of YRNAs, especially YRNA1, correlated with a vast number of proteins and cellular processes associated with viral infections and immunologic responses to viruses. HNSCC-derived cell lines overexpressing YRNA1 were then used to determine the correlation of YRNA1 and the expression of genes associated with HPV infections. Taken together, our results highlight the potential of YRNAs as possible HNSCC biomarkers and new molecular targets.

## 1. Introduction

Head and neck squamous cell carcinomas (HNSCCs) are among the most challenging tumor types to treat [1,2,3]. They originate from epithelial cells of the aerodigestive tract [2,3,4] and may be classified according to their localization: nasopharyngeal, tongue, oral, and laryngeal squamous cell carcinoma (NSCC, TSCC, OSCC, and LSCC, respectively) [2,3,5,6]. The most common risk factors are alcohol consumption, tobacco smoking, and human papilloma virus (HPV) infection [7,8]. Interestingly, among younger patients, HPV infection is the most important risk factor, often associated with better treatment outcomes and recovery [1,2,3,4,6,9,10]. Over 200 types of HPV have been described, but HPV-16 is the strain mostly associated with squamous cell carcinoma [10,11].

HPV(+) and HPV(−) tumors molecularly differ from each other. The mutation rate in HPV(+) HNSCC is lower compared with HPV(−) HNSCC, and HPV(+) tumors are characterized by a lower number of *TP53* mutations [10,11]. Clinically, HPV(+) cells are more radiosensitive. HPV(+) HNSCC has altered mismatch repair systems, DNA repair mechanisms, and homologous recombination pathways, which significantly contribute to HNSCC cells’ radiosensitivity. Moreover, another important factor in HPV(+) radiosensitivity is the overexpression of p16, which delays the DNA damage response [10].

HPV(−) and HPV(+) HNSCC development includes epigenetic changes including regulatory RNAs, which take part in various crucial processes such as apoptosis regulation, proliferation, cell migration, and cell cycle regulation [1,2,6]. However, these epigenetic changes differ between HPV(−) and HPV(+) subtypes which are manifested in different potential of HNSCC aggressiveness [1,2,6]. 

Non-coding RNA molecules (ncRNAs) are essential for many cellular processes [1,2,4,9,10,12,13,14,15,16]. Previous studies have demonstrated that many different types of ncRNAs, including long non-coding RNAs (lncRNA) and short non-coding RNAs such as miRNA, are dysregulated in HNSCC [1,3,4,9,10,12,17,18,19]. Moreover, ncRNAs may be used as very promising biomarkers and targets for future molecular-based therapies [1,3,4,9,10,12,17,18,19].

One of the studied types of ncRNAs are YRNAs (Ro associated-Y). These ncRNAs are components of Ro60 ribonucleoprotein particles and consist of 80–112 nucleotides [1,12,13,20,21,22,23]. Four different YRNAs may be distinguished: YRNA1, YRNA3, YRNA4 and YRNA5 [1,12,13,20,21,22,23]. These four YRNA genes are clustered at a single chromosomal locus on chromosome 7q36 and are transcribed by RNA polymerase III [1,12,13,14,20]. Mature YRNAs form a stem-loop structure. The upper stem of YRNAs is essential for the initiation of DNA replication, leading to the formation of new replication forks. The lower stem is a Ro60-binding site, forming an activated protein–ribonucleoprotein complex [1,12,13,20,21]. The lower stem also controls the nuclear export of YRNAs. YRNAs have been shown to interact with many different proteins conditioning their functions [12,13,20,24]. YRNAs also were found in extracellular vesicles and in retroviruses [12,13,20,24].

Since YRNAs are easily obtained from human serum, plasma, saliva, and tissues, it makes them potential biomarkers and targets for future therapies [12]. Studies have shown that YRNAs are over-expressed in glioma [25], triple-negative breast cancer (TNBC) [26], pancreatic ductal adenocarcinoma (PDAC) [27,28], colon cancer [29,30], cervix cancer [27,29], benign prostate hyperplasia [31] and clear-cell renal-cell carcinoma (ccRCC) [14]. On the other hand, YRNAs have been found to be downregulated in HNSCC [1], prostate cancer [15], and bladder cancer [16]. YRNAs are naïvely involved in crucial processes of cancer development such as apoptosis, cell proliferation, angiogenesis, metastasis, and different types of cellular stresses [12]. However, the involvement of YRNAs in viral-associated tumors, such as HPV infections in HNSCC, is unknown and their biological role is not yet defined. In order to address this role, YRNAs’ (YRNA1, YRNA3, YRNA4, and YRNA5) expression patterns were investigated in publicly available RNAseq datasets of HPV(−) and HPV(+) HNSCC tissue samples. Additionally, HNSCC-derived cell lines overexpressing YRNA1 were used to determine the correlation between YRNA1 and the expression of genes associated with HPV infections. The main aim of this study was to show the impact that YRNAs have on HNSCC development as well as to show their correlation with HPV infection, which is an important HNSCC development factor.

## 2. Materials and Methods

### 2.1. HNSCC Gene Expression Datasets

For the expression analysis of YRNA patterns in HNSCC, data generated by the Leipzig Head and Neck Group (LNHG) for 269 HNSCC cases (including 196 HPV(−) and 73 HPV(+) cases) were used [31]. They comprised expression data of 31330 genes as specified by Illumina IDs (HumanHT-12_V4_0_R2_15002873_B). The expression data were obtained from patients’ biopsies. The patients were treated with different therapeutic approaches according to tumor subsites and TNM stages. The corresponding metadata are available in association with the gene expression data at the following GEO accession ID: GSE65858. The specific information considering raw data and processing were described in detail by Wichmann et al. [31].

Additionally, for the validation of expression values following overexpression of YRNA1 in FaDu and Detroit cell lines, data from 523 HNSCC samples obtained from cBioPortal (Head and Neck Squamous Cell Carcinoma, TCGA, PanCancer Atlas) were used. All data are available and access is unrestricted. The data derived from GEO were used to calculate the expression of YRNA1 in terms of clinical parameters and perform GSEA analysis. The genes from the GEO and TCGA datasets were also used for the validation of NGS results.

### 2.2. YRNAs Expression and Clinical Parameters

The correlation between each member of the YRNA cluster (YRNA1, YRNA3, YRNA4, and YRNA5) and clinical parameters were determined and associated with HPV status. 

The following clinical parameters were considered: age (below or above 60 years), smoking status (Yes vs. No/Ex; Ex—ex smoker), tumor localization (oropharynx, larynx, hypopharynx, oral cavity), HPV16 infection status (as measured by p16 positive vs. negative), TP53 mutation status (WT vs. disruptive vs. non-disruptive), T stage (T1 + T2 vs. T3 + T4 and T1 vs. T2 vs. T3 vs. T4), N stage (N0 + N1 vs N2 + N3 and N1 vs. N2 vs. N3 vs. N4), clusters (classical vs. basal vs. atypical vs. mesenchymal), clinical stage (I + II vs. III + IV and I vs. II vs. III vs. IV) and HPV(−) and HPV(+) status. 

The expression levels of YRNA1, YRNA3, YRNA4, and YRNA5 were also associated with active and inactive viral status (DNA + RNA+ vs. DNA + RNA-) and type of HPV (HPV16 vs. other types of HPV) in the HPV(+) group. 

The association of YRNA1 with specified cancer markers was also calculated. The cancer markers such as *CD44*, *SOX2*, *TP53*, *ALDH1A1*, and *FAT1* were chosen because of their vast influence on various processes occurring in HNSCC as shown in previously published papers [32,33,34,35,36,37,38,39,40,41].

Overall survival (OS) and progression-free survival (PFS) analyses were performed using two subgroups (low and high expression of YRNAs, based on the mean expression levels) in HPV(+), HPV(−), and both together. The subgroups were compared using Log-Rank (Mantel–Cox), Gehan–Breslow–Wilcoxon, and Hazard Ratio (Mantel–Haenszel; HR) tests. The 95% Confidence Interval (CI) of the ratio was calculated.

Finally, the ROC analysis was applied to compare YRNA1, YRNA3, YRNA4, and YRNA5 expression levels between HPV(−) and HPV(+), and AUC (Area under the ROC curve) were calculated.

### 2.3. Functional Enrichment Analysis and Prediction of Gene Function

Gene Set Enrichment Analysis (GSEA) software version 4.1.0 (http://www.gsea-msigdb.org/gsea/index.jsp, accessed on 15 February 2022) was used as previously described for the analysis of functional enrichment [42,43]. The input file (GEO accession ID: GSE65858) contained expression data for 29,089 genes and 269 patients. HPV(−) and HPV(+) groups were divided into high- and low-expression subgroups based on the mean expression levels of YRNA1, YRNA3, YRNA4, and YRNA5.

Analysis of the Oncogenic Signatures (C), Hallmark Gene Set (H), and Gene Ontology (GO) with 1000 gene set permutations were applied and a nominal *p*-value *p* ≤ 0.05 and FDR *q*-value ≤ 0.25 were considered statistically significant. Next, using the Reactome database, genes derived from GSEA were analyzed in terms of pathways in human organisms (http://reactome.org, accessed on 8 March 2022) [44]. Finally, the interactions between protein-coding genes in the pathway which were the most significantly enriched in a group of patients with low vs. high expression of YRNAs were analyzed using the GeneMANIA prediction tool (http://genemania.org) [45]. The expression heat map was generated using Morpheus Heat Map (https://software.broadinstitute.org/morpheus/, accessed on 15 March 2022). 

### 2.4. Estimation of Immune Cells Fractions

The deconvolution method was applied to analyze the immune cell composition in HNSCC tissues based on expressional data from patients samples. This technique was performed using a tool developed by Chiu et al. [46], the source code of which is based on commonly available R and Python packages and can be downloaded from https://github.com/holiday01/deconvolution-to-estimate-immune-cell-subsets (accessed on 20 June 2022). All steps of the deconvolution analysis were carried out in line with the authors instructions [46]. To correctly prepare the GEO data set for this analysis, the list of gene names contained therein was updated in accordance with the HGNC nomenclature [47], and, subsequently, any repetitions resulting from the presence of expression data for different transcripts of the same gene were removed. Patients were divided based on the YRNA1 expression level. The ten individuals with the highest and ten with the lowest levels of studied YRNA formed two groups which were then compared. High- and low-expression groups were extracted on the same basis from the general population of patients (*n* = 269), as well as separately from subgroups with HPV(−) (*n* = 196) and HPV(+) status (*n* = 73). The deconvolution provided estimated fractions of 9 immune cell types, including naïve CD4 T cells, natural killer cells, macrophages M1 and M2, dendritic cells, T helper cells, regulatory T cells, naïve CD8 T cells, as well as memory CD8 T cells for each individual, which were then compared and statistically evaluated.

### 2.5. Cell Line, Transfection, RNA Isolation, and qRT-PCR

Two HNSCC-derived cell lines were used for YRNA1 gain-of-function assays, FaDu and Detroit562. The FaDu cell line was cultured as previously described [48], and Detroit562 was cultured in DMEM (Biowest, Nuaille, France) with 10% FBS (Biowest, Nuaille, France) and geneticin antibiotic (KRKA, Novo Mesto, Slovenia). The cell lines were examined for mycoplasma using the VenorGeM Mycoplasma PCR Detection Kit (Minerva Biolabs, Berlin, Germany). Cell lines were seeded on 6-well plates (400,000 cells/well, 5% CO_2_ and 37 °C) and transfected using pcDNA3.1(+)-hRNY1[NR_004391.1] or with pcDNA3.1(+)-CTR (control) vectors obtained from VectorBuilder Inc. (VectorBuilder Inc., Chicago, IL, USA) with Lipofectamine with PLUS Reagent (Thermo Scientific, Waltham, MA, USA) according to the manufacturer’s protocol. After 48 h incubation, the transfected cells were selected using G418 antibiotic (0.8 mg/mL for FaDu and 0.4 mg/mL for Detroit562) for another 7 days at 5% CO_2_ and 37 °C.

Total RNA from the cell lines was isolated using a Total RNA Midi isolation kit (A&A Biotechnology, Gdańsk, Poland), according to the isolation protocol. Next, the quality and quantity of isolated RNA samples were examined using the NanoDrop 2000 spectrophotometer (Thermo Scientific, Waltham, MA, USA), followed by 28S and 18S rRNA band estimation (1% agarose gel electrophoresis in TAE (Tris-acetate-EDTA (Ethylenediaminetetraacetic acid buffer). The enhanced expressions of the YRNA1 in FaDu and Detroit562 cell lines were confirmed with qRT-PCR. Complementary DNA was synthesized using an iScript cDNA Synthesis Kit (Bio-Rad, Hercules, CA, USA), and 0.5 μg of the total RNA was used. Quantitative PCR was performed using 2× concentrated SYBR Green Master Mix (Roche, Basel, Switzerland) with specific primers to detect YRNA1 as described previously [15,30,31]. Endogenous control *HPRT1* (F: 5′-TGA CCT TGA TTT ATT TTG CAT ACC-3′ AND R: 5′-CGA GCA AGA CGT TCA GTC CT-3′) was used at a final reaction concentration of 0.5 μM with 5× diluted cDNA. The real-time PCR reactions were performed on a LightCycler 96 (Roche, Basel, Switzerland) device, and a melting curve was performed to discriminate between non-specific products of the PCR reaction. All real-time PCR data were analyzed by calculating the 2^−ΔCT^, normalizing against the mean of *HPRT1* expression.

### 2.6. RNA Sequencing

Sequencing of RNA samples derived from FaDu and Detroit562 cells overexpressing YRNA1 and controls was performed by Eurofins Genomics Europe Sequencing GmbH (Eurofins Scientific, Luxembourg) using the Genome Sequencer Illumina NovaSeq and NovaSeq 6000 S4 PE150 XP. Human Genome hg19/GRC37, UCSC; annotations Gencode v29, Ensembl 90 were used as references of annotation. 

Briefly, high-quality sequence reads were aligned to the reference genome using STAR (Spliced Transcripts Alignment to a Reference), run through the Sentieon framework along with the known gene models [49]. The STAR algorithm achieves highly efficient mapping by performing a two-step process: seed searching, followed by clustering, stitching, and scoring. The percent of mapped transcripts for FaDu_pcDNA3.1(+)-hRNY1[NR_004391.1], FaDu_pcDNA3.1(+)-CTR, Detroit562_pcDNA3.1(+)-hRNY1[NR_004391.1] and Detroit562_pcDNA3.1(+)-CTR to a reference genome was 95.3%, 92.2%, 98.7% and 98.4%, respectively. Next, gene quantification was achieved by inspecting transcriptome alignment using the RSEM tool [50]. Read counts were further normalized to account for sequencing depth and gene length biases. Fragment per kilobase per million (FPKM) and transcripts per million (TPM) values were generated. Finally, differentially expressed gene identification was performed. To identify a gene or a transcript differentially expressed, Cuffdiff 2 tests the observed log-fold-change in expression against the null hypothesis of no change. Because measurement error, technical variability, and cross-replicate biological variability might result in an observed log-fold-change that is not zero, Cuffdiff 2 assesses significance using a model of variability in the log-fold-change under the null hypothesis. This method is described in detail by Trapnell et al. [51].

### 2.7. Statistical Analysis

All statistical analyses were performed using GraphPad Prism 9 (San Diego, CA, USA). The normality of the groups was tested using the Shapiro–Wilk test and subsequent comparisons of the two groups were carried out using the *t*-test or Mann–Whitney U test depending on the distributions. For the comparison of three and more groups, one-way ANOVA, Kruskal–Wallis test, and post-tests Dunn’s multiple comparison test or Tukey’s multiple comparison test were used. The correlation analysis between YRNAs and gene markers was performed using the Spearman correlation test. The REACTOME pathway browser was used as a free tool for pathway analysis of genes derived from NGS (www.reactome.org, accessed on 1 February 2023). In all analyses, *p* < 0.05 and FDR < 0.25 was considered significant.

## 3. Results

### 3.1. The Expression of YRNAs Is Significantly Distinct between HNSCC Clinical and Pathological Stages

First of all, the data obtained from the GEO dataset were examined in terms of the expression of YRNA1, YRNA3, YRNA4, and YRNA5 in association with different clinical–pathological features. The analyses of YRNA1 showed that YRNA1 was highly dysregulated in terms of the tumor’s N stage. It was noticed that the expression of YRNA1 was significantly higher in N2–N3 stages (*p* = 0.0426), and the same trend was observed after clustering N stages into two groups: N0 + N1 vs. N2 + N3 (*p* = 0.0110) (Figure 1A). YRNA3 was only significantly overexpressed in T3–T4 stages (*p* = 0.0045), with the highest expression seen in the T3 stage. When clustering T stages into two groups (T1 + T2 vs. T3 + T4) the highest expression of YRNA3 was confirmed in T3 + T4 stages (*p* = 0.0019) (Figure 1A). The rest of the YRNAs did not show any significant changes in this case. In the case of the clinical stage, the expression of YRNA3 also showed significant changes (*p* = 0.002) (Figure 1A). Interestingly, further clustering the data into two groups (I + II vs. III + IV) showed that YRNA3 was significantly overexpressed in the III–IV clinical stages (*p* = 0.0002). Interestingly, all four YRNAs showed significant alterations in their expression levels among different subtypes (*p* < 0.0001, *p* = 0.0006, *p* = 0.0051, *p* = 0.0318, respectively) (Figure 1). It was noticed that lower expression levels of YRNAs were found in the least aggressive HNSCC subtypes such as classical and basal subtypes, and the expression increased in the mesenchymal subtype—the most aggressive one. These data show that the expression of YRNAs is correlated with the advancement of the HNSCC disease. Finally, the YRNAs did not show any significant changes in their expression in terms of tumor localization (Figure 1A). All values for all cases may be found in Appendix A. 

The ROC analysis was applied and AUC (Area under the ROC curve) was calculated to compare HPV(+) vs. HPV(−), DNA + RNA+ vs. DNA + RNA-, and p16 vs. other HPV; however, no significant differences were observed (Appendix A).

Since previous studies showed that YRNAs may be associated with HPV infection, the data were divided into two groups: HPV(+) and HPV(−). In the HPV(−) group, YRNA1 was found to be significantly upregulated in N2 + N3 compared with N0 + N1 (*p* = 0.0193). YRNA3 was significantly upregulated in both T3 and T4 stages (*p* = 0.0110) in the HPV(−) group and when T3 + T4 stages were grouped (*p* = 0.0003) (Figure 1B). A similar trend was seen for the clinical stage analysis, where the expression of YRNA3 was upregulated in the III and IV clinical stages, both separately or when taken together (*p* = 0.0204 and *p* = 0.0022, respectively). Interestingly, YRNA5 showed the opposite results. The expression of YRNA5 was found to be overexpressed in both T1 and T2 stages (*p* = 0.0437) and in T1 + T2 stages (*p* = 0.0430) (Figure 1B). YRNA5 was also found to be overexpressed in clustered I + II clinical stages (*p* = 0.0470) (Figure 1B). Next, the YRNAs’ expression was analyzed in different HNSCC subtypes: classical (the least aggressive subtype), basal, atypical, and mesenchymal (the most aggressive subtype). Lower expression of YRNAs is associated with the least-aggressive tumor subtype and higher expression was associated with the more-aggressive subtype (*p* < 0.0001 for YRNA1; *p* = 0.0224 for YRNA4; *p* = 0.01 for YRNA5) (Figure 1B). YRNA3 did not show any differences in expression levels in this case. Additionally, no differences in expression levels of YRNAs were found in terms of the N stage (except for the clustered analysis of YRNA1) and tumor localization (Figure 1B). All values for these analyses are presented in Appendix A.

In HPV(+) group, which showed better survival rates and better treatment outcomes among HNSCC patients, it was discovered that the expression of YRNA3 is significantly overexpressed in III and IV clinical stages (*p* = 0.0449) and the clustered III + IV variant (*p* = 0.0330) (Figure 1C). Similar results were observed in the analysis of all patients and the HPV-negative group.

Surprisingly, it was observed that YRNA3 is significantly overexpressed in the hypopharynx in comparison with the larynx (*p* = 0.0434) (Figure 1C). Moreover, in terms of expression in different HNSCC subtypes, YRNA4 and YRNA5 did not show any differences; however, in the case of YRNA1 and YRNA3, a similar trend may be seen as with that in the previous analysis (*p* = 0.0075 and *p* = 0.0053, respectively) (Figure 1C). In both cases, it was noticed that a higher expression of YRNA1 and YRNA3 correlates with more-aggressive tumor subtypes. The rest of YRNAs did not show any expression differences in terms of localization. There were also no differences in terms of the T stage and N stage in the HPV(+) group. All values for these analyses are presented in Appendix A.

### 3.2. YRNAs Have a Distinct Impact on Cancer and Stemness Markers

Next, correlations of YRNAs and *CD44*, *SOX2*, *TP53*, *ALDH1A1* and *FAT1* cancer and stemness markers were analyzed in a whole group of patients and divided into atypical, basal, classical and mesenchymal subtypes.

In the whole group of patients, YRNA1 showed a significant negative correlation with *CD44* (ρ = −0.1716; *p* = 0.0051), *SOX2* (ρ = −0.2707; *p* < 0.0001), *ALDH1A1* (ρ = −0.1421; *p* = 0.0206), and *FAT1* (ρ = −0.2501; *p* < 0.0001) (Figure 2A). The correlation with *TP53* was also slightly negative; however, it did not show any statistical significance.

YRNA4 was found to be negatively correlated with *SOX2* and *ALDH1A1* (ρ = −0.3009, *p* < 0.0001; ρ = −0.3096, *p* < 0.0001, respectively). YRNA5 was also negatively correlated with *SOX2* (ρ = −0.1962, *p* = 0.0013) but also with *FAT1* (ρ = −0.1995, *p* = 0.0011). YRNA3 was not significantly correlated with any of the examined cancer and stemness markers (Figure 2A).

Next, it was found that the correlation of YRNA1 and cancer and stemness markers vastly differed between the most and the least aggressive subtypes. In the most aggressive subtype, mesenchymal, the correlation of YRNA1 and the selected markers was negative for all examined markers, albeit statistically significant in the case of *SOX2* and *TP53* (ρ = −0.2916, *p* = 0.0075; ρ = −0.2462, *p* = 0.0249, respectively). In the mesenchymal subtype, YRNA3 showed a significant positive correlation with *CD44* (ρ = 0.3832, *p* = 0.0003) (Figure 2B). On the other hand, in the classical subtype, which is known to be the least aggressive HNSCC subtype, it was noticed that most correlations were slightly positive and in one case there was a statistical significance: *TP53* and YRNA4 were significantly correlated with each other (ρ = 0.4833, *p* = 0.0079) (Figure 2B). In the atypical subtype, YRNA1 was negatively correlated with all markers, but only correlation with *SOX2* showed statistical significance (ρ = −0.2595, *p* = 0.0277). YRNA4 was significantly, negatively correlated with *CD44*, *SOX2*, and *ALDH1A1* (ρ = −0.2764, *p* = 0.0817; ρ = −0.3177, *p* = 0.0065; ρ = −0.4578, *p* < 0.0001, respectively), and YRNA5 was found to be negatively correlated with *CD44* (ρ = −0.233, *p* = 0.0489) (Figure 2B). Finally, in the basal subtype, YRNA1 was negatively correlated with all markers and significance was seen in *CD44*, *SOX2*, and *FAT1* (ρ = −0.248, *p* = 0.0256; ρ = −0.2799, *p* = 0.0114; ρ = −0.3119, *p* = 0.0046, respectively). A significant, positive correlation was observed only between YRNA3 and *CD44* (ρ = 0.2957, *p* = 0.0074). Negative correlations were also observed between YRNA4, *SOX2* and YRNA5, *FAT1* (ρ = −0.2722, *p* = 0.014; ρ = −0.2755, *p* = 0.0128, respectively) (Figure 2B). 

### 3.3. Patients’ Overall Survival and Progression-Free Survival Are Associated with YRNAs Expression Levels

The associations between patients’ survival rates and expression levels of YRNAs, which is the most important clinical–pathological parameter, were analyzed. Patients were divided into two groups and the mean value of YRNA expression levels were used as a cut-off for all patients, HPV(+) and HPV(−). It was observed that patients with low expressions of YRNA4 showed better survival than patients in the high-expression groups (*p* = 0.002, HR = 0.4332, 95% CI = 0.2782 to 0.6744) during 2500 days of observation. For the rest of the analyzed YRNAs, no differences were observed (*p* > 0.05) (Figure 3A). However, when the time of observation was shortened to 1000 days, patients with lower expressions of YRNA1 and YRNA4 showed significantly better overall survival rates (*p* = 0.0024, HR = 0.4727, 95% CI = 0.2916 to 0.7664 and *p* = 0.0002, HR = 0.3926, 95% CI = 0.2382 to 0.6470, respectively) (Appendix A).

Next, patients were divided based on HPV status and overall survival depending on YRNAs level was analyzed (Figure 3B). No differences in patients’ overall survival depending on YRNA3, YRNA4, and YRNA5 levels in the HPV(+) group were noticed (*p* > 0.05). However, patients with higher levels of this gene had a longer overall survival time than the group of patients with lower expression levels of YRNA1 (*p* = 0.0451, HR = 2.559, 95% CI = 1.021 to 6.416) (Figure 3B). However, in the shortened time none of YRNAs showed any significant differences in overall survival of patients (Appendix A).

In the HPV(−) group, patients with lower levels of YRNA1 and YRNA4 had longer survival times during 2500 days of observation (*p* = 0.0036, HR = 0.4914, 95% CI = 0.3046 to 0.7928 and *p* < 0.0001, HR = 0.2880, 95% CI = 0.1750 to 0.4741, respectively) and when time was shortened to 1000 days (*p* = 0.0008, HR = 0.4053, 95% CI = 0.2386 to 0.6886 and *p* < 0.0001, HR = 0.3016, 95% CI = 0.1748 to 0.5203, respectively) than those with higher levels of those genes. However, no differences (*p* > 0.05) between patients’ overall survival and YRNA3 as well as YRNA5 were noticed during 2500 and 1000 days of observation (Figure 3B and Appendix A).

The analysis of patients’ progression-free survival is described in Figure 4. Considering the whole group of HNSCC patients, those with low expressions of YRNA4 showed better progression-free survival than patients in the high-expression group (*p* = 0.0034, HR = 0.5764, 95% CI = 0.3985 to 0.8338). No differences (*p* > 0.05) between YRNA1, YRNA3, and YRNA5 levels and patients’ progression-free survivals were observed (Figure 4A). However, when shortening the time of observation to 1000 days, significantly longer progression-free time was noticed for patients with lower expressions of YRNA1 and YRNA4 (*p* = 0.0167, HR = 0.6377, 95% CI = 0.4411 to 0.9218 and *p* = 0.0062, HR = 0.5801, 95% CI = 0.3927 to 0.8569, respectively). YRNA3 and YRNA5 did not show any significant differences when shortening the observation time (Appendix A).

In the HPV(+) group, only patients with higher expression levels of YRNA1 showed longer progression-free survival rates (*p* = 0.0373, HR = 2.180, 95% CI = 1.047 to 4.540). No differences for YRNA3, YRNA4, and YRNA5 were observed (*p* > 0.05) (Figure 4B). In spite of the shortened time of observation to 1000 days, no differences for YRNAs were indicated (Appendix A).

In the HPV(−) group, patients with lower expression levels of YRNA1 and YRNA4 showed significantly better outcomes than those with higher expression levels of those genes (*p* = 0.0293, HR = 0.6401, 95% CI = 0.4285 to 0.9560 and *p* < 0.0001, HR = 0.4258, 95% CI = 0.2793 to 0.6492, respectively) (Figure 4B). No differences in survival for YRNA3 and YRNA5 were observed (*p* > 0.05). Similar results were found in a shortened analysis time to 1000 days. Patients with lower expressions of YRNA1 and YRNA4 showed significantly better progression-free survival rates than patients with higher expressions of those genes (*p* = 0.0246, HR = 0.6162, 95% CI = 0.4041 to 0.9398 and *p* = 0.0021, HR = 0.5067, 95% CI = 0.3287 to 0.7812, respectively). YRNA3 and YRNA5 did not show any significant differences (Appendix A).

### 3.4. YRNAs Are Correlated with Different Genes among the HPV(+) Group with an Influence on HPV Proteins and Viral and Immunologic Pathways

The YRNAs were correlated with all genes derived from the examined GEO dataset in two groups: HPV(+) and HPV(−) (Figure 5A, Appendix A). The Venn diagrams in Appendix A depict gene distribution in YRNAs between HPV(+) and HPV(−) groups (Appendix A). Interestingly, there were 21 common genes for YRNA1, 15 genes for YRNA3, 7 genes for in YRNA4, and 8 common genes for YRNA5 observed between the HPV(+) and HPV(−) groups. In the HPV(+) group, YRNA1 was negatively correlated with 25 genes and positively correlated with 7 genes. YRNA3 was negatively correlated with 16 genes and positively correlated with 9 different genes. In the case of YRNA4, 6 genes were negatively correlated and 9 genes were positively correlated. YRNA5 was negatively correlated with 14 genes and positively correlated with 6 genes (Figure 5A). 

Next, the genes correlated with YRNAs were found to take part in many viral and immunologic processes, such as antigen processing and presenting, regulation of the innate immune system, and different cellular responses (Figure 5B). Moreover, a closer examination of protein-coding genes correlated with YRNA1 showed that many of them were strictly correlated with HPV proteins (Figure 5C). Three genes were correlated with E1 protein, three with E2 protein, six with E5 protein, three with E6 protein, ten with E7 protein, two with L1 protein, and one with L2 protein (Figure 5C). These HPV proteins are essential for HPV infection and stability. All of them were correlated with a vast number of different viral and immunologic processes associated with HPV invasion and replication in cells (Figure 5D).

### 3.5. YRNA1 Significantly Correlates with Protein Secretion Processes

Gene Set Enrichment Analysis (GSEA) was performed to obtain functional implications of YRNA1 expression in the HPV(+) groups. Interestingly, both groups showed the same outcome. The highest-enriched pathway correlated with YRNA1 was the protein secretion pathway (Normalised Enrichment Score—NES—1.5823789) (Figure 6A). Next, the interactions between protein-coding genes in the protein secretion pathway were analyzed using the GeneMANIA prediction tool (Figure 6B). The analysis showed 16 genes that are mostly co-expressed with each other (40.96%). These genes were strictly correlated with the expression of YRNA1 (Figure 6A). Moreover, those 16 significantly altered genes were further analyzed using the REACTOME pathway browser, resulting in annotation of the proteins of those genes to specific processes in the human organism, such as the immune system, signal transduction, cell–cell communication, cellular stress to external stimuli, transport of small molecules, vesicle-mediated transport, metabolism of proteins, developmental biology, metabolism, and different diseases (Figure 6C). Finally, a created heat map shows that patients with higher-expressed YRNA1 have, in most cases, lower-expressed examined protein-coding genes and the lower-expressed group showed correlations with the higher-expressed protein-coding genes (Figure 6D).

In the GSEA analysis for YRNA3, YRNA4, and YRNA5 many more pathways were enriched than in YRNA1 (Figure 7). In the HPV(−) group YRNA3 was enriched in 20 different pathways and in the HPV(+) group in 11 different pathways. Worth noting is that 10 of these pathways were positively enriched and only one pathway showed a negative NES value—JAK2 DN, which is a pathway connected with different genes that are downregulated after the JAK2 downregulation. Next, YRNA4 in the HPV(+) group was found to be enriched in 59 pathways (22 most important shown on the figure); however, in the HPV(−) group YRNA4 was not enriched in any of the examined pathways. Many of these pathways are associated with DNA repair, ribosomes, and mitochondria. YRNA4 is also enriched in such crucial processes for YRNAs’ functions as SNRNP (small nuclear ribonucleoprotein) assembly, ribonucleoprotein complex subunit organization, and ribonucleoprotein complex biogenesis. Finally, the YRNA5 in the HPV(+) group was enriched in five pathways, but none in the HPV(−). These pathways and genes involved in them such as TGF beta signaling are essential in cancer development. All the NES, nominal *p* values, and FDR q values may be found in Appendix A.

### 3.6. YRNA1 Expression Significantly Correlates with Immune Cells

All HNSCC patients, as well as HPV(+) and HPV(−) subgroups, were divided into low- and high-YRNA1-expression groups, and immune cell content in patients’ tumor samples was predicted using deconvolution analysis. In the case of all patients, a significantly larger amount of DC cells was found in the groups showing lower expressions of YRNA1. Similar results were obtained in the HPV(−) group in addition to significantly higher amounts of M2 macrophages in the group with high expressions of YRNA1. In HPV(+) there were no significant differences discovered; however, the trend for DC cells was maintained (Figure 8).

### 3.7. Overexpressed YRNA1 Upregulates Genes Associated with Responses to Viral Infection

The overexpression of YRNA1 in FaDu and Detroit562 cells was confirmed by qRT-PCR analysis, with cells transfected with pcDNA3.1(+)_hRNY1 expressing significantly higher levels of YRNA1 than in cell lines transfected with pcDNA3.1(+) (*p* = 0.0002 and *p* = 0.0008, respectively) (Figure 9A).

Analysis of RNA sequencing data of FaDu and Detroit562 cells overexpressing YRNA1 using the REACTOME pathway browser showed enrichment of viral infection-associated pathways (infectious disease pathway, influenza infection pathway, viral mRNA translation, and influenza viral mRNA transcription, FDR < 0.25 and *p* < 0.05). For the Detroit562 cell line, additional alterations were seen for host interactions, activation/modulation of innate and adaptive immune responses, modulation of host translation machinery, and targeting host intracellular signaling and regulatory pathways (FDR < 0.25 and *p* < 0.05) (Figure 9A). 

Furthermore, the genes selected from FaDu and Detroit562 cell lines overexpressing YRNA1 were examined using the GeneMANIA prediction tool, which allows the prediction offunctions and pathways of genes. The genes derived from the modified FaDu cell line were mostly associated with viral gene expression and viral latency. However, these genes were also involved in immune responses—regulating cell surface, receptor-signaling pathway involved in phagocytosis, Fc-gamma receptor signaling pathway, Fc-receptor mediated stimulatory signaling pathway, ribosome biogenesis, and cotranslational protein targeting the membrane. These results also showed that 59.01% of these genes were co-expressed (Figure 9B). The genes derived from the modified Detroit562 cell line are also involved in viral gene expression; however, many of them are involved in immunological processes such as antigen binding, antigen processing, and presentation of endogenous antigen and peptide antigen and the MHC protein complex. In this case, 68% of the examined genes were found to be co-expressed (Figure 9B).

Interestingly, 40 genes out of the 100 most-abundant genes following the overexpression of YRNA1 were common for modified FaDu and Detroit562 cell lines from which 23 showed similar expression patterns and 17 of them showed opposite expression patterns between the examined cell lines (Figure 9C). Further analysis using the REACTOME pathway browser also allowed distinguishing 40 genes from the modified FaDu cell line and 38 from the modified Detroit562 cell line to be involved in processes that include infectious diseases, the life cycles of SARS-CoV viruses, influenza virus and HIV, some metabolic processes mediated by intracellular *Mycobacterium tuberculosis*, the actions of clostridial, anthrax, diphtheria toxins, and the entry of *Listeria monocytogenes* into human cells (Figure 9D, Appendix A).

A subset of genes selected from FaDu and Detroit562-overexpressing YRNA1 were validated on the TCGA and GEO datasets, including the group of 40 genes expressed in both cell lines (mentioned above) and those expressed in only one cell line, constituting a set of 60 other genes. From the second dataset, eight genes in each group were not analyzed due to different factors such as inaccuracy in gene nomenclature between the datasets or the absence of some genes in the TCGA or GEO databases. Nevertheless, the validation of NGS data allowed us to discover that 18 out of 40 common genes were significantly dysregulated in data obtained from the TCGA database. In the case of FaDu, 22 out of 52 genes were dysregulated and in the Detroit562 cell line 24 genes out of 52 were significantly dysregulated (Figure 10). Furthermore, validation on the GEO database showed that 4 genes out of 20 common genes showed significant changes between the HPV(+) and HPV(−) groups. In genes specific only for the FaDu cell line, 7 genes out of 52 were significantly dysregulated and in Detroit562 it was 11 genes out of 52 (Figure 10). 

## 4. Discussion

Head and neck squamous cell carcinomas still lack successful treatments because of their high aggressiveness and molecular heterogeneity [1,4,9]. It is crucial to find new approaches not only for the treatment but also for the detection of the disease in its early stages. One of the main factors for HNSCC development is HPV infection status, showing differences not only in the expression of genes but also in the treatment outcome. It has been previously shown that HPV infection is associated with younger patients and HPV-positive patients show better survival rates [4,9,52,53]. Interestingly, most HPV(+) HNSCCs are histologically graded as very poorly differentiated tumors in spite of their better clinical outcome, and in general less-differentiated tumors tend to show more aggressive behavior [54]. Additionally, HPV(+) HNSCC tumors differ from HPV(−) in immune and mutational profiles and in gene expression [6]. For example, in HPV(+) HNSCCs, *TP53* is rarely changed because p53 is eliminated by the action of the E6 HPV protein. In HPV(−) tumors, this gene is not eliminated and usually, it is mutated [6]. Moreover, in HPV(+) HNSCC tumors the E7 HPV protein binds to RB1—a cell cycle regulator—retinoblastoma-associated protein and causes its proteasomal degradation. The lack of RB1 causes the release of E2F family transcription factors, which results in cells skipping the G1-S checkpoint and going straight into the S phase [6]. In recent years, many researchers focused on looking for new therapeutic targets or biomarkers for HNSCC [1,4,9,12] based on the measurement of proteins, DNA, and RNA levels [1,2,4,9,10,11,12,13,14,25,26], including non-coding RNA (ncRNA) molecules [1,12,13,14,25,26].

One of such ncRNAs is YRNA, which plays important roles in many processes correlated with tumor development [1,12,13,14,25,26]. YRNAs were also previously found to bind with many different proteins determining their different functions in an organism [12]. There are four distinguished types of YRNAs: YRNA1, YRNA3, YRNA4, and YRNA5. All of the YRNAs are transcribed by RNA polymerase III. YRNAs characterize a stem-loop structure and every part of a YRNA is responsible for different processes [1,12,13,14,25,26]. Previous studies have proved that YRNAs may be abundantly found in extracellular vesicles and retroviruses where they function as scaffolds for viral RNA [20]. Lately, YRNAs were discovered to be easily obtained from different body fluids and tissues. They are dysregulated in many cancer types, including HNSCC [1,12,14,15,16]. 

In this work, we analyzed data from 269 patients in terms of expression levels of YRNAs YRNA1, YRNA3, YRNA4, and YRNA5, in HPV(−) and HPV(+) groups to answer if they have any biological function and could be used as potential biomarkers depending on HPV status.

First of all, the expressions of YRNAs were analyzed in the context of clinical–pathological parameters. It was observed that the overall survival and progression-free survival analyses showed very similar results; in the general group as well as in the HPV(−) group, better survival rates were observed in the low-expression group and in the HPV(+) group better survival rates were seen in the high-expression group. These results correlate with the results considering the expression in different disease stages shows that higher expression of YRNAs is correlated, in most cases, with more advanced cancer stages, and with more aggressive HNSCC subtypes. Despite the time of observation used in a particular analysis, YRNA4 and YRNA1 showed, in most cases, potential for improving patients’ survival biomarkers, especially in terms of the HPV(+) and HPV(−) groups, because of opposite results. However, OS and PFS in terms of YRNAs differ in different cancer types. In another study considering OS and PFS in HNSCC, higher YRNA1 expression showed better survival, contrastingly to these results. However, this difference may occur because of a lack of information considering HPV status in our previous study [1]. In clear-cell renal carcinoma, better survival rates for YRNA3 were observed for lower expression, but for higher expression in terms of YRNA4 [14]. In the case of prostate cancer, YRNA5 showed better survival rates in low-expression groups [15], and in bladder cancer, all YRNAs showed better survival rates in higher-expression groups [16]. These studies show how YRNAs differ between different cancer types not only in terms of expression but also patients’ survival. This may suggest developing different approaches for YRNAs in different cancers.

YRNA3 is correlated with more advanced T-stages of the disease, making it a promising biomarker of the HNSCC progression. Interestingly, in our previously published study, an analysis based on FFPET tissues from HNSCC patients showed that YRNA3 was the only YRNA that did not show any statistical significance in terms of the T-stage [1]. This difference may be connected with different sample types used in both analyses. Furthermore, high YRNA3 expression was found to be correlated with more advanced clinical stages despite HPV infection status, showing that YRNA3 may function as a potential biomarker of more advanced stages of the disease. On the other hand, in a different study concerning the clear-cell renal-cell carcinoma, YRNA3 was discovered to be overexpressed in less-advanced cancer stages (I + II vs. III + IV clinical stage) [14]. Such differences may occur because of different tumor types. Previous studies have shown high sensitivity and specificity of YRNA3 as a potential biomarker in bladder cancer [16], prostate cancer [15], clear-cell renal-cell carcinoma [14], pancreatic ductal adenocarcinoma [12], and HNSCC [1]. These results suggest YRNA3 as a potential biomarker of different diseases in the future. Furthermore, it was discovered that YRNA3 may also function as a distinguishing biomarker between the larynx and hypopharynx in the HPV-negative group. Unfortunately, the rest of the YRNAs may not be used as biomarkers of the HNSCC localization because of the lack of significant differences in expression between different localizations. 

YRNA1 showed significant differences in expression patterns only in terms of N-stage when all patients were taken together. It was noticed that higher expression of YRNA1 is correlated with more advanced N-stages of the disease. This may suggest that YRNA1 may be also used as a biomarker of early metastasis to nearby lymph nodes. However, in other studies, YRNA1 showed significantly dysregulated expression patterns. In a previous study, YRNA1 was overexpressed in more advanced T-stages and because of its high specificity and sensitivity, it was predicted as a possible biomarker of HNSCC [1]. It was also found that the expression of YRNA1 is significantly lower in tumor tissue compared with adjacent healthy tissue [1]. Similarly to YRNA3, YRNA1 was examined in other studies concerning different tumor types, resulting in the association of YRNA1 as a possible biomarker of bladder cancer [16], prostate cancer [15], pancreatic ductal adenocarcinoma [12], and cervix cancer [27,29]. Interestingly, YRNA1 was found to be under-expressed in bladder cancer [16] and prostate cancer [15], but overexpressed in cervix cancer [27,29] and pancreatic ductal adenocarcinoma [12]. Such differences between various types of tumors may suggest that YRNA1 may be not only a great biomarker but also show the importance of its role in developing different tumors. 

Next, the analysis showed that YRNA5 was only significantly dysregulated in the HPV(−) group, suggesting that it may function as a biomarker of HPV infection, similarly to YRNA1 [1]. Interestingly, the expression of YRNA5 was upregulated in the less-advanced T-stage and clinical stage of the disease. This may also suggest its value as a biomarker of less-advanced tumor stages. YRNA4 and YRNA5 were previously described by us to be overexpressed in FFPET samples of HNSCC patients [1]. In clear-cell renal-cell carcinoma YRNA4 was found to be significantly overexpressed in kidney tissue compared with healthy tissue. Its higher expression was also indicated in less-advanced N-stages of the disease and patients with low expression of YRNA4 showed better survival rates [14]. In bladder cancer, both YRNA4 and YRNA5 were found to be significantly downregulated, and thus they were proposed as disease and progression biomarkers [16]. Similarly, YRNA4 and YRNA5 were significantly under-expressed, suggesting their potential as biomarkers in prostate cancer. In benign prostate hyperplasia both of these YRNAs were submitted as biomarkers because of their significant overexpression [15]. What is more, in colon cancer, YRNA4 was highly expressed in the blood serum of rectal cancer patients [12]. All these data show how different expression patterns of YRNAs present in various diseases, pointing to their future potential as molecular targets or functional biomarkers.

The expression of YRNAs was compared in terms of the aggressiveness of the HNSCC. It was discovered that higher expression of YRNAs is correlated with more aggressive tumor subtypes. The highest expression was found in the mesenchymal subtype, which is known to be the most aggressive HNSCC subtype [31]. As the aggressiveness changes the correlation and probably also a function of YRNAs in HNSCC development also change, suggesting that YRNAs may play different roles depending on the advancement and aggressiveness of the tumor, suggesting that YRNAs have a great impact on developing, growing and metastasizing tumors. However, more studies need to be carried out on this matter to discover the exact mechanism and function in each case. Unfortunately, knowledge concerning the role of YRNAs in HPV infection and HNSCC development is still very limited.

The five most common cancer and stemness markers *CD44*, *SOX2*, *TP53*, *ALDH1A1* and *FAT1* were analyzed in terms of YRNAs and their potential influence on each other [32,33,34,35,36,37,38,39,40,41]. *CD44* is a cancer stem cell marker associated with cell aggregation, proliferation and migration [32]. It is also partially responsible for HNSCC invasion and poor survival of patients. It may also be used as a poor prognosis indicator in HNSCC [33]. *SOX2* is one of key regulators in HNSCC and takes part in cancer stemness. Moreover, it is correlated with oral squamous cell carcinoma metastasis [34,35]. Next, *TP53* is the most common altered gene in HNSCC. It is altered in approximately 70% of cases [36]. *TP53* is responsible for the activation of DNA repair mechanisms and apoptosis induction due to DNA damage [37]. Another marker, *ALDH1A1*, is not expressed in the normal oral mucosa. It functions in carcinogenesis and tumor progression of HNSCC and is associated with poorer prognosis [38,39]. *FAT1* is mutated in approximately 20% cases [40], and is associated with tumor progression and survival of HNSCC patients [41]. It was found that YRNAs were mostly negatively correlated with these cancer and stemness markers. YRNA1 was significantly, negatively correlated with *CD44*, *SOX2*, *ALDH1A1* and *FAT1*. It is well known that these genes have a huge impact on tumor development and maintenance of cancer stem cells and could be used as potential biomarkers [32,33,34,35,36,37,38,39,40,41]. In the classical subtype, which is the least aggressive subtype, mostly positive correlations between YRNAs and chosen cancer and stemness markers may be seen; however, in the mesenchymal the opposite results were obtained. These results imply a huge impact of YRNAs on cancer progression in subtypes with different aggressiveness. 

Furthermore, the Gene Set Enrichment Analysis revealed that YRNA1 expression is strongly associated with the protein secretion pathway. Interestingly, the HPV(−) and HPV(+) group showed the same results with the same altered protein coding genes. It was found that 16 significantly dysregulated protein-coding genes were associated with the YRNA1 low-expression group. These genes are involved in many crucial processes such as metabolism, developmental biology, disease, the immune system, metabolism of proteins, vesicle-mediated transport, transport of small molecules, cellular response to external stimuli, cell–cell communication and signal transduction. Many of these protein coding genes were previously described in different cancer diseases including HNSCC and some of them are correlated with HPV infection. The first of these genes, AP-2 complex subunit mu (*AP2M1*) was found to be a promising biomarker for predicting survival of patients with hepatocellular carcinoma [55]. The *AP2M1* is an important factor in hepatitis C virus (HCV) assembly. Moreover, it is also associated with low-risk HPV6 and high-risk HPV16 by binding to the E7 proteins of the HPV [56]. In HNSCC it is discovered to be one of the most significant predictors of disease-free survival and overall survival and is upregulated in more-advanced cancer stages [57]. The correlation of *AP2M1* and YRNA1 may explain the potential role of YRNA1 as a HPV infection indicator [1]. AP-1 complex subunit gamma-1 (*AP1G1*) takes part in developing colon cancer [47] and in liver cancer [58]. In HNSCC, *AP1G1* was found to be significantly overexpressed. The knockdown of *AP1G1* results in indirect sensitization of HNSCC cells to cetuximab and possibly increases the therapeutic outcomes of HNSCC treatment [59]. Brefeldin A-inhibited guanine nucleotide-exchange protein 1 (*ARFGEF1*) was found to be downregulated in breast cancer [60]. It also takes part in papillary thyroid cancer proliferation, migration and invasion [61]. In colon cancer cells, *ARFGEF1* is one of the targets of *miR-27b* in regulating cell proliferation. *miR-27b* downregulates *ARFGEF1*, leading to tumor growth suppression [62]. This suggests that YRNA1 may be involved in tumor growth in colon cancer. Moreover, in Kaposi’s Sarcoma induced by KSHV (Kaposi’s sarcoma-associated herpes virus) circulating *ARFGEF1* was found to be significantly overexpressed and associated with induced cell migration, proliferation and angiogenesis [63]. Next, Coatomer subunit beta (*COPB2*) was downregulated in cervical cancer [64] and upregulated in breast cancer [65]. It is associated with colorectal cancer cell proliferation and apoptosis [66]. The *COPB2* protein was also related with SARS-CoV-2 virus [67]; however, there are no findings considering *COPB2* in HPV infection. Copper-transporting ATPase 1 (*ATP7A*) is highly expressed in esophageal squamous cell carcinoma [68] and is correlated with tumorigenesis and cisplatin resistance [69]. AP-3 complex subunit beta-1 (*AP3B1*) is upregulated in hepatocellular carcinoma [70] and is a proven target for *miR-9* in breast cancer cells [71]. Moreover, together with *AP3S1*, it is downregulated in cervical cancer [72]. Next, *YIPF6* is significantly overexpressed in castration-resistant prostate cancer [73]; however, in the same disease *BET1*’s lower expression is associated with early relapse [74]. In addition, *BET1* is also associated with better prognosis in glioblastoma [75]. Ras-related protein Rab-5A (*RAB5A*) was previously indicated to be overexpressed in oral cancer [76], cervical cancer tissue [77] and in colorectal cancer [78]. Interestingly, *RAB5A* is essential for the induction of autophagy by HCV (Hepatitis C Virus). In terms of HPV infection, HPV16 virions colocalize with RAB5A-containing components. *RAB5A* is crucial for biogenesis and coordination of endosomes and autophagosomes which would suggest that virions may transit through autophagosomes [53]. Furthermore, protein MON2 homolog (*MON2*) regulated by *microRNA-133a-5p* inhibits metastatic capacity of clear-cell renal carcinoma [79]. Moreover, *MON2* is required for efficient production of infectious HIV-1 particles [80]. Finally, Vacuolar protein sorting-associated protein 4B (*VPS4B*) was found to be downregulated in rectum adenocarcinoma, colon adenocarcinoma, ovarian serous cystadenocarcinoma, adrenocortical cancer and testicular germ cell tumor [81]. On the other hand, high expression of *VPS4B* is associated with faster cell proliferation and poor prognosis in hepatocellular carcinoma [82]. Additionally, dominant negative mutants of *VPS4A* and *VPS4B* inhibit the replication and release of Hepatitis B virus [83]. There is little known in terms of HPV infection and cancer diseases in terms of *DST*, *OCRL* and *STAM*. Our results and previous studies have shown huge potential of YRNA1 in regulating various protein-coding genes in different cancer types, which may be used in developing new targeted therapies. Additionally, through interaction between YRNA1 and proteins such as *AP2M1* and *RAB5A* we can implicate that YRNA1 has actually a vast impact on HPV infection.

The GSEA analysis of YRNA3, YRNA4 and YRNA5 showed many more processes that these YRNAs are enriched in, and the most important processes for YRNA3 in the HPV(−) group are extracellular transport and mitosis G2M transition checkpoint, both of them being important in cancer genesis. Worth noticing is that YRNA3 was negatively correlated in these processes, suggesting that the downregulation of YRNA3 may have a positive impact on tumor regression. YRNA3 in HPV(+) was mostly positively correlated with many processes; however, in one case it was negatively enriched in the JAK2 signaling pathway which plays a central role in cytokine and growth factor signaling [84]. Furthermore, enrichment in genes implicated in DNA repair processes, ribosome assembly processes, ribonucleoprotein complex biogenesis, spliceosomal SNRNP (small nuclear ribonucleoproteins) assembly and ribonucleoprotein complex subunit organization was observed in the case of YRNA4, which suggest its crucial role in forming a Ro60 ribonucleoprotein complex, one of the basic functions of YRNAs [1,2,12,13,25]. Finally, YRNA5 in the HPV(+) group was enriched in WNT beta catenin signaling and TGF beta signaling, crucial pathways in cancer development [85,86]. This would suggest that YRNA5 is indirectly responsible for epithelial–mesenchymal Transition in HNSCC development through the WNT beta catenin signaling pathway. We can also conclude a connection between YRNA5, HPV infection and YRNA5’s influence on the EMT process in HNSCC, which results in tumor progression and metastasis. However, more studies are needed to fully understand this mechanism. All these GSEA data show that YRNAs play more important roles in cancer genesis than was suggested before [1].

To thoroughly analyze the topic, the deconvolution analysis discovered that YRNA1 is associated with immune cells, especially dendritic cells. These cells are responsible for antigen uptake and presentation to activate and regulate anti-tumor T cell response [87]. In our analysis, a higher amount of these cells was associated with low expression of YRNA1 and low expression of YRNA1 was found in HNSCC tissue and cell lines. Our previous studies indicated that patients with high levels of YRNA1 survive longer periods of time and YRNA1 expression is very low in HNSCC [1]. Taking all of the abovementioned into consideration, it can be assumed that YRNA1 may function as a tumor suppressor for HNSCC.

Next, the analysis of correlation of YRNAs and different genes in HPV(+) and HPV(−) groups showed that despite the group, YRNAs were vastly positively and negatively correlated with different genes. Most of those genes were previously correlated with different types of tumors [88,89,90,91,92,93,94,95,96,97,98]. Interestingly, some of the genes were common for different YRNAs such as *SNX17*, *WDR6* or *ATP5B*, suggesting that different YRNAs have an impact on similar genes. These results underline how important the role of YRNAs is in developing different types of cancers.

Finally, the RNAseq of FaDu and Detroit562 cells overexpressing YRNA1 compared with control cell lines showed us that YRNA1 plays a role in the cellular response to viral infections. Out of the 100 most-abundant genes derived from the NGS analysis, more than half of them were associated with viral processes such as host interaction of HIV factors, integration of provirus, infectious disease, influenza infection, influenza viral RNA transcription and replication and viral mRNA translation. These results formed the basis for a further look at whether YRNA1, as well as other YRNAs, due to their similar homology, function and shared promoter, perform a function in HPV HNSCC infection.

It should be noted that we decided to use HPV-negative cell lines in our in vitro model due to eliminating the transcription changes caused by viral infection and observed only changes made by YRNA1. Moreover, pcDNA3.1 plasmid, used by us in this model, should not induce viral response effects in the cell such as viral particles generated for cell modification in lentiviral systems. However, we are aware of the simplicity of the presented model and it most certainly only partially shows the importance of YRNAs in viral infection.

Based on RNAseq results, it was observed that out of the 100 most-abundant genes in both examined cell lines, 40 genes were found in FaDu as well as in Detroit562. Among these genes, 17 of them had opposite expression trends between the cell lines and 23 showed similar ones. These differences may occur due slightly different collection sites of the cell lines at the beginning; the FaDu cell line was obtained from solid, primary hypopharyngeal tumors and Detroit562 cells were obtained from lymph node metastasis of pharyngeal cancer patients. Previous studies already indicated differences between solid primary tumor cells and lymph node metastasis cells not only in gene expression [99] but also in the mechanics and structures of these cells [100]. However, in both cell lines, changes in genes connected with response to viral and other infections was observed. The GeneMANIA prediction tool allowed us to confirm genes derived from NGS analysis to be involved in viral gene expression, viral latency and immune response. Previous studies also showed that YRNA5-derived fragments are responsible for inhibition of influenza virus infection [101]. Another study also showed a number of proteins interacting with different YRNAs that are involved in various viral infections [24]. Moreover, the genes derived from NGS analysis were validated on a GEO dataset and a TCGA dataset, confirming their abundance in HNSCC patients. The deeper analysis of protein-coding genes correlated with YRNAs, especially YRNA1, showed that 28 of these genes are strictly correlated with HPV proteins and additionally correlated with other genes involved with YRNA1 expression. YRNA1 was also found to be correlated with many immune processes such as antigen presenting and processing, regulation of the innate immune system, different cellular responses and many more. These findings only confirm the vast influence of YRNA1 on different viral infection types, including HPV infection in different tumor types.

Despite the very promising results concerning the role of YRNAs in HNSCC and viral infections, there is still a lot to discover. More studies are needed to fully understand the YRNAs interactions and their influence on different cancer types. In this study, despite no significant difference in YRNA1 expression between HPV status groups, we found much more data confirming the correlation with HPV infection. Interestingly in previous studies, a significant correlation between YRNA1 and HPV status were found [1,12]. Such differences in YRNAs expression may occur due to different extraction sites of specimen (plasma, serum, tissue, biopsy, FFPET) [102]. There are also no studies concerning the influence of different therapeutic agents (chemotherapy, radiotherapy) on YRNA expression. In our preliminary data we can confirm at this point that radiotherapy and chemotherapy cause significant changes in YRNA1 expression. Similar results between YRNAs may occur because of their conservative structure and similar functions. In this study we focused on YRNA1; however, as the results suggest it would be beneficial to study YRNA3, YRNA4 and YRNA5 in the future as well. For now, YRNA1 shows properties of HNSCC biomarkers and correlates with HPV infection and immune response to that infection. YRNA3, YRNA4 and YRNA5 also show properties of potential biomarkers of the disease itself, as well as the prognosis for HNSCC patients. Taken together all YRNAs showed properties to be promising molecular targets for future therapies, not only for virus-induced cancers but also for other diseases. 

## 5. Conclusions

In this study, YRNAs in terms of their influence on HNSCC development and HPV infection were examined. First of all, YRNA1 and YRNA3 were associated with more-advanced cancer stages, and YRNA5 was associated with less-advanced cancer stages, suggesting a potential role of these YRNAs as biomarkers for HNSCC tumors. Next, we found that the higher the expression of YRNAs the more aggressive tumor subtype. Additionally, YRNAs were associated with cancer and stemness markers showing their negative correlation between them, and opposite correlations between the most and the least aggressive subtypes, showing a distinct impact of YRNAs on HNSCC depending on the aggressiveness of the tumor and the HNSCC development. It was also discovered that YRNA1 and YRNA4 may be potential prognostic biomarkers of survival, differing between HPV(+) and HPV(−) groups of patients. Next, YRNA1 was found to be correlated with HPV infection and immune response to cancer disease. The results showed a significant correlation of YRNA1 and HPV proteins and immune processes. On the other hand, YRNA5 was found to be overexpressed only in the HPV(−) group, making it a potential biomarker on HPV infection status in HNSCC. YRNAs also were found to be enriched in a vast number of processes correlated with cancer genesis and viral and immunogenic pathways. The overexpression of YRNA1 in HNSCC-derived cell lines confirmed the expression of genes co-expressed with YRNA1, and suggest a role for YRNA1 in viral infections, including HPV infection in HNSCC patients. All these findings show how YRNAs may interfere in cancer progression, especially in association with HPV infection, and should be evaluated as biomarkers and potential therapeutic targets.

## Figures and Tables

**Figure 1 biomedicines-11-00681-f001:**
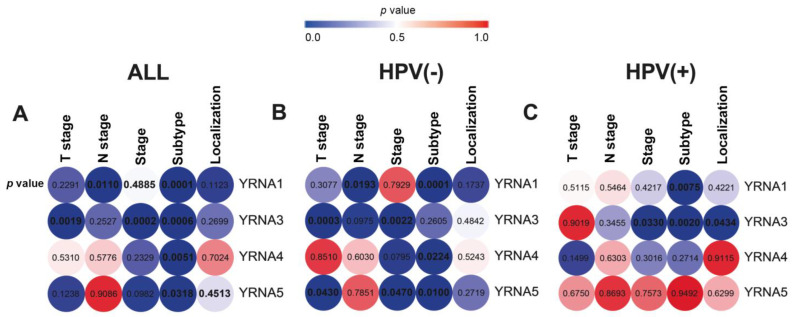
(**A**) GEO dataset expression levels of YRNA1, YRNA3, YRNA4, and YRNA5 in terms of T stage (T1 + T2 vs. T3 + T4), N stage (N0 + N1 vs. N2 + N3), clinical stage (I + II vs. III + IV), HNSCC subtypes and localization of (**A**) all HNSCC patients (*n* = 269); (**B**) HPV negative (*n* = 196) and (**C**) HPV positive (*n* = 73). Shapiro–Wilk normality test, T-test or Mann Whitney Test; One-WAY ANOVA, Post test: Dunn’s Multiple Comparison Test or Tukey’s Multiple Comparison Test; the graphs show relative expression and mean of value with SEM; *p* < 0.05 considered significant.

**Figure 2 biomedicines-11-00681-f002:**
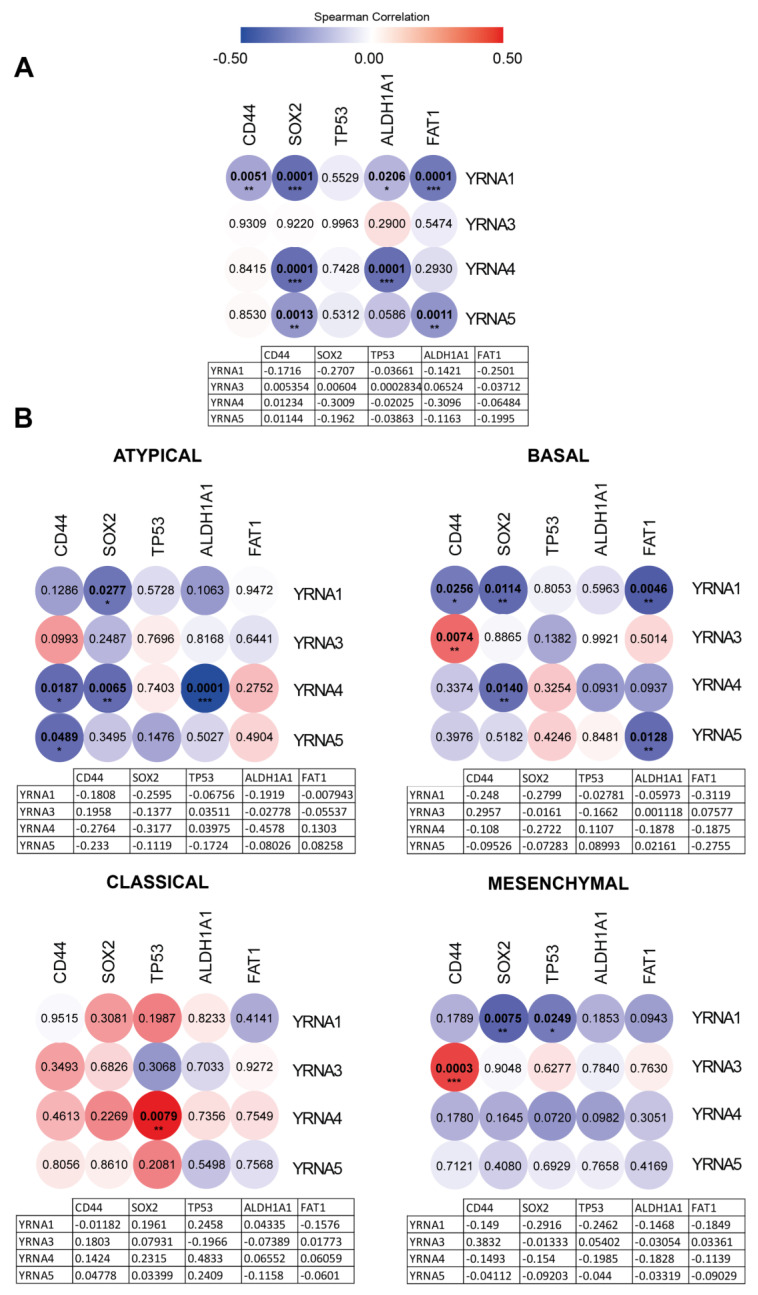
Correlograms of Spearman correlation between YRNAs and chosen cancer and stemness markers based on the GEO dataset: (**A**) for the whole analyzed group (*n* = 265); (**B**) for different HNSCC subtypes: atypical (*n* = 72), basal (*n* = 81), classical (*n* = 29) and mesenchymal (*n* = 83). Spearman correlation, *p* < 0.05 considered significant; * *p* < 0.05; ** *p* < 0.01; *** *p* < 0.001.

**Figure 3 biomedicines-11-00681-f003:**
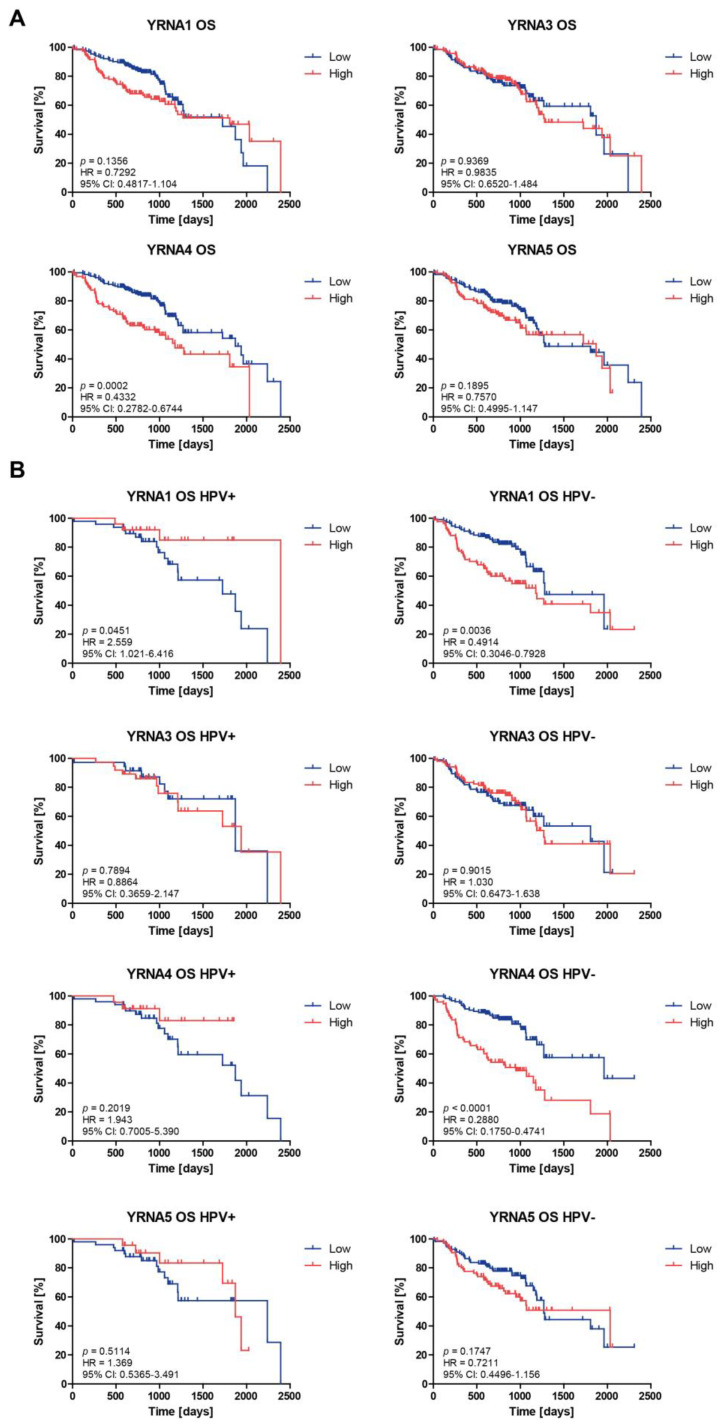
Association between the expression levels of YRNA1, YRNA3, YRNA4, and YRNA5, and HNSCC patients’ overall survival based on the GEO dataset: (**A**) whole group of patients (HPV(−) and HPV(+)); (**B**) those in only the HPV(+) and in only the HPV(−) groups. Low- and high-expression groups of patients were divided based on mean as a cut-off calculated separately for every one of the groups, all patients (YRNA1: low *n* = 152, high *n* = 117; YRNA3: low *n* = 128, high *n* = 142; YRNA4: low *n* = 174, high *n* = 96; YRNA5 low *n* = 164, high *n* = 106), HPV(+) (YRNA1: low *n* = 48, high *n* = 25; YRNA3: low *n* = 36, high *n* = 37; YRNA4: low *n* = 50, high *n* = 23; YRNA5 low *n* = 50, high *n* = 23), HPV(−) (YRNA1: low *n* = 112, high *n* = 84; YRNA3: low *n* = 94, high *n* = 102; YRNA4: low *n* = 123, high *n* = 73; YRNA5 low *n* = 111, high *n* = 85).

**Figure 4 biomedicines-11-00681-f004:**
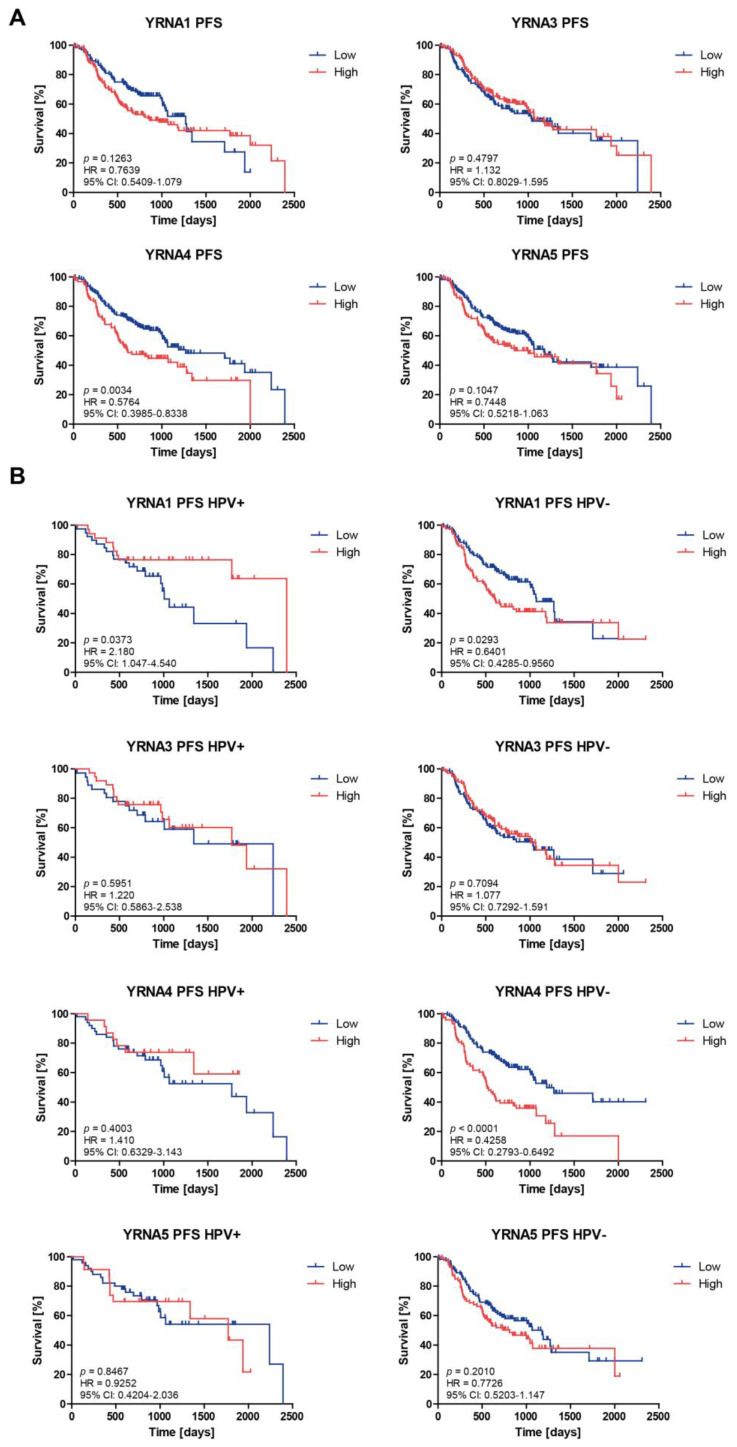
Association between expression level of YRNA1, YRNA3, YRNA4, and YRNA5, and HNSCC patients’ progression-free survival based on the GEO dataset: (**A**) whole group of patients (HPV(−) and HPV(+)); (**B**) in only the HPV(+) group and in only the HPV(−) group. Low- and high-expression groups of patients were divided based on mean as a cut-off calculated separately for every one of the groups: all patients (YRNA1: low *n* = 132, high *n* = 138; YRNA3: low *n* = 128, high *n* = 142; YRNA4: low *n* = 174, high *n* = 96; YRNA5 low *n* = 164, high *n* = 106), HPV(+) (YRNA1: low *n* = 39, high *n* = 34; YRNA3: low *n* = 36, high *n* = 37; YRNA4: low *n* = 50, high *n* = 23; YRNA5 low *n* = 50, high *n* = 23), HPV(−) (YRNA1: low *n* = 112, high *n* = 84; YRNA3: low *n* = 94, high *n* = 102; YRNA4: low *n* = 123, high *n* = 73; YRNA5 low *n* = 110, high *n* = 86).

**Figure 5 biomedicines-11-00681-f005:**
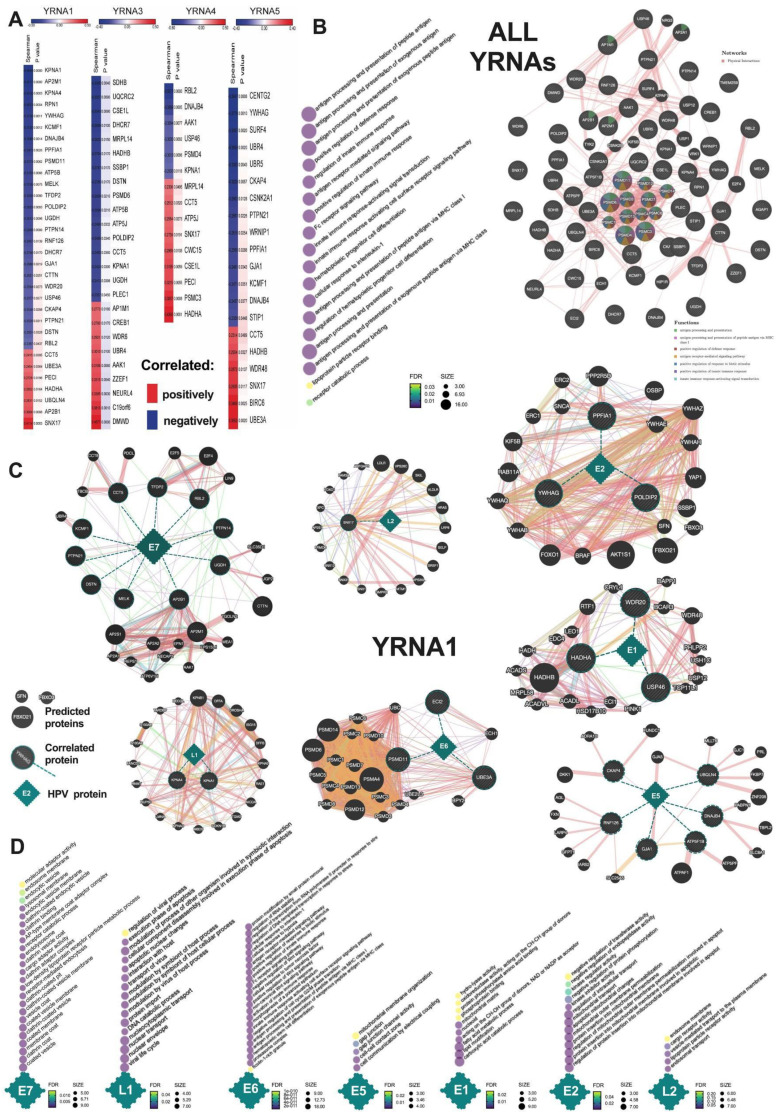
Correlation analysis of YRNAs with genes from the GEO dataset of HPV(+) HNSCC patients in terms of HPV infection: (**A**) heat maps of correlated genes between YRNAs and different genes involved in HPV infection; (**B**) functional analysis of changed genes and their involvement to biological processes; (**C**) association of YRNA1’s correlated genes with HPV proteins and (**D**) involvement of correlated and predicted genes in biological processes depending on HPV proteins. Spearman correlation, *p* < 0.05 considered significant; analysis based on the GeneMANIA tool, FDR < 0.1.

**Figure 6 biomedicines-11-00681-f006:**
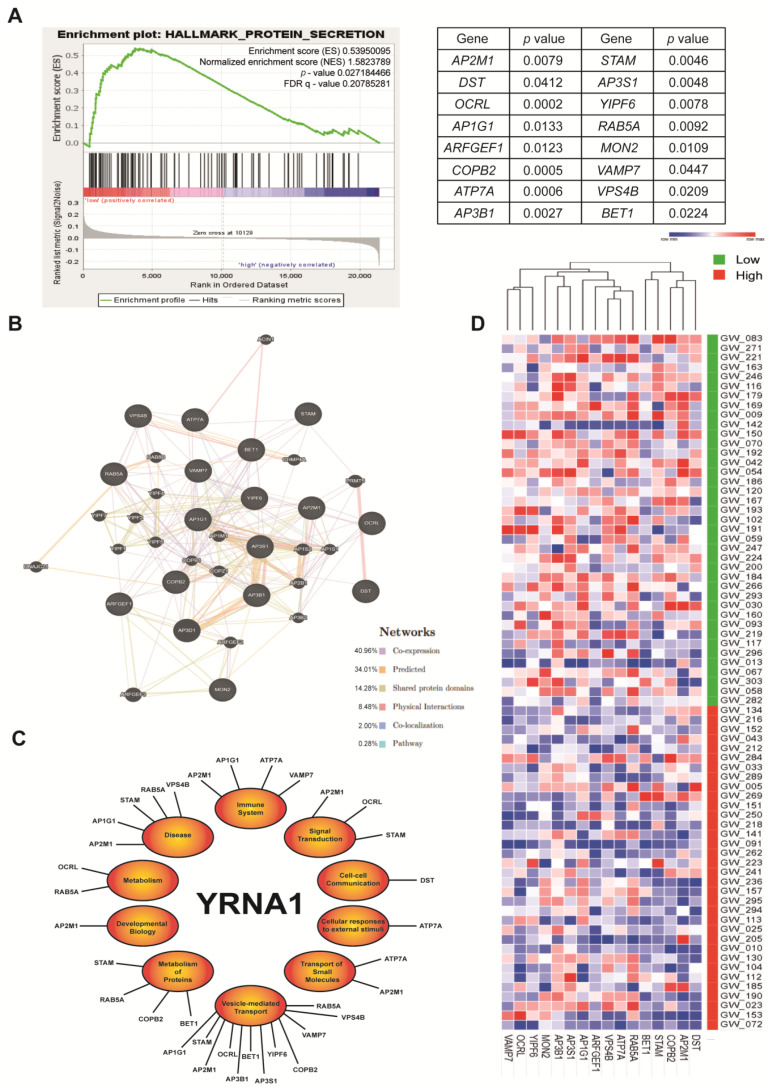
(**A**) Changed genes in the HPV-positive group of patients with higher expressions of YRNA1; (**B**) GeneMANIA analysis; (**C**) a diagram of functional genes correlated with YRNA1 based on the REACTOME pathway browser; (**D**) the expression of correlated protein-coding genes with YRNA1 obtained from GSEA; Mann–Whitney Test; *p* < 0.05 considered significant.

**Figure 7 biomedicines-11-00681-f007:**
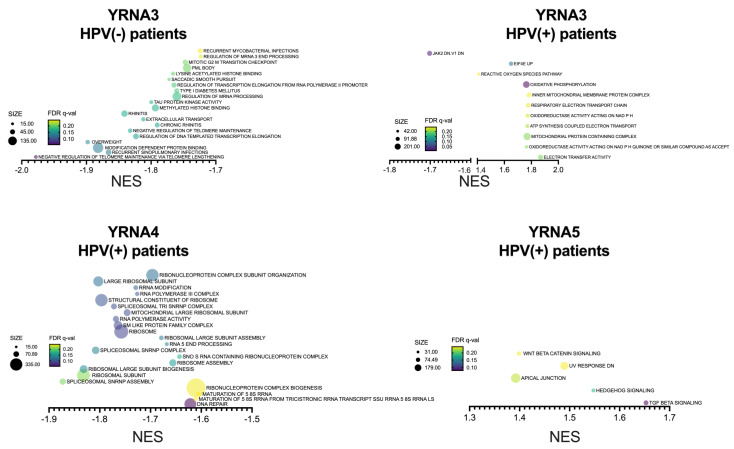
The Gene Set Enrichment Analysis of YRNA3, YRNA4, and YRNA5 in the HPV(+) and HPV(−) groups; nominal *p*-value *p* ≤ 0.05 and FDR *q*-value ≤ 0.25 considered as significant.

**Figure 8 biomedicines-11-00681-f008:**
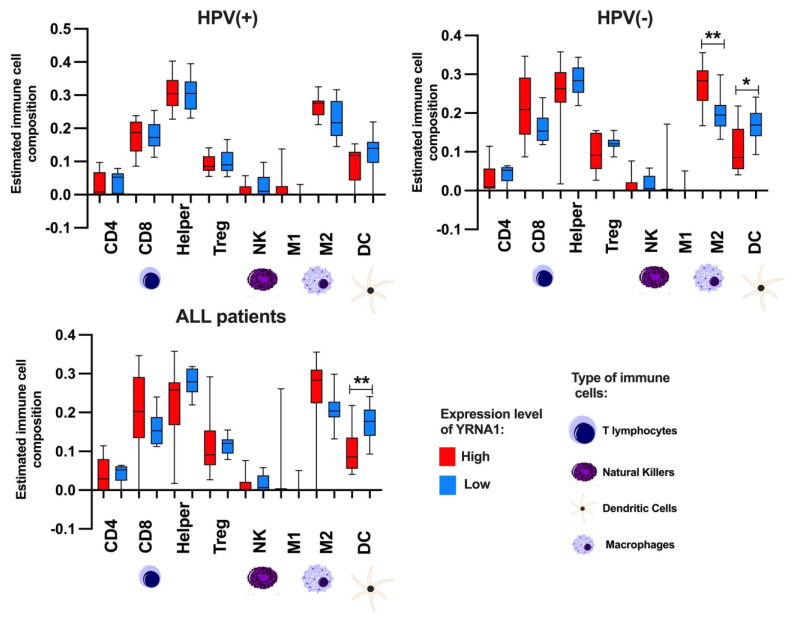
The deconvolution analysis of immune cells depending on high and low expression levels of YRNA1 in the group of only HPV(+), only HPV(−) and all patients; *p* < 0.05 considered significant, * *p* < 0.05; ** *p* < 0.01.

**Figure 9 biomedicines-11-00681-f009:**
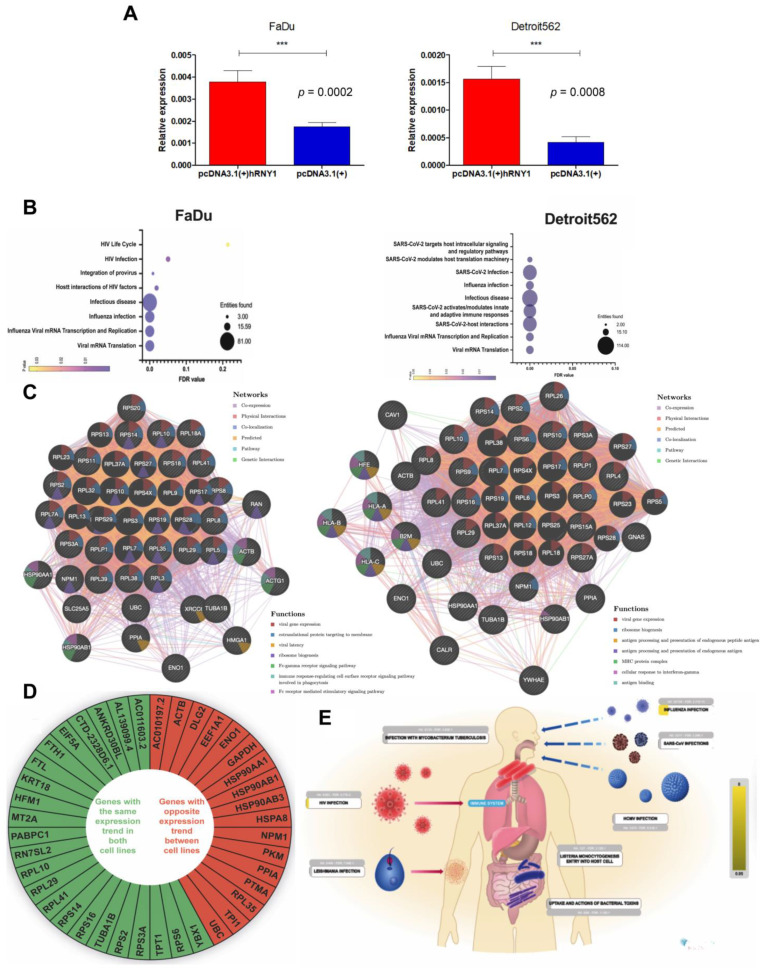
The results of gene expression analysis of FaDu and Detroit562 cell lines overexpressing YRNA1. (**A**) YRNA1 expression in FaDu and Detroit562 cell lines after transfection with pcDNA3.1(+)_hRNY1 and pcDNA3.1(+). Shapiro–Wilk normality test, Mann–Whitney Test; the graphs show relative expression and mean of value with SEM; *p* < 0.05 considered significant, *** *p* < 0.001; (**B**) pathways in which the most abundant genes from FaDu_hRNY1 and Detroit562_hRNY1 cell lines were associated using the REACTOME pathway browser; (**C**) GeneMANIA prediction tool analysis of genes derived from YRNA1-overexpressing FaDu and Detroit562 cell lines after NGS and REACTOME analysis; (**D**) a diagram showing common genes between FaDu and Detroit562-overexpressing YRNA1 cell lines derived from NGS; (**E**) scheme depicting different viral infections correlated with overexpressed YRNA1; *p*-value ≤ 0.05 and FDR *q*-value ≤ 0.25 considered significant.

**Figure 10 biomedicines-11-00681-f010:**
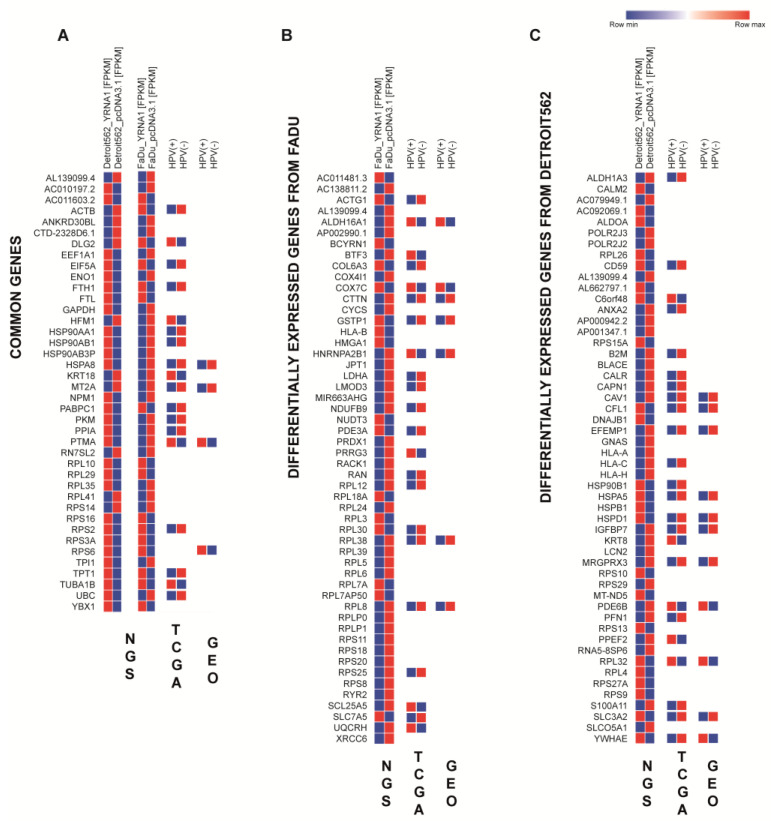
Validation of NGS results using TCGA data and a GEO data set. (**A**) Validation of common genes derived from the NGS analysis for FaDu and Detroit-overexpressing YRNA1; (**B**) validation of genes specific for FaDu cell line overexpressing YRNA1 derived from NGS analysis; (**C**) validation of genes specific for Detroit562 cell lines overexpressing YRNA1 derived from NGS analysis. Shapiro–Wilk normality test, *t*-test or Mann–Whitney test; *p* < 0.05 considered significant.

## Data Availability

The data presented in this study are openly available at GEO and TCGA databases, and its analysis and results presented in this work, does not violate any copyrights.

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
