# Peer review of "The Impact of YRNAs on HNSCC and HPV Infection"

_biomedicines, 2023, doi:10.3390/biomedicines11030681_

Round 1

Reviewer 1 Report (Previous Reviewer 2)

In this re-submission from Guglas and colleagues, the authors present a revised study regarding noncoding yRNAs in HNSCC samples and cell lines and a putative connection to HPV infections. They analyzed total RNAseq data from cell lines overexpressing Y1 yRNA as well as a publicly available HNSCC microarray dataset and a polyA-enriched RNA seq HNSCC dataset from the TCGA project.

Major concerns:

  • Almost all analyses the authors mention are still described inappropriately. For some analyses it is not clear why the authors performed them at all.

  • All results sections merely list some numbers without any conclusions or connections to the results of other subsections.

  • The introduction and Discussion sections still contain inappropriately many self-citations. E.g., when the authors introduce yRNAs, they list 2 of their own review papers, however, they do not mention important original works, e.g. Hendrick et al (PMID: 6180298) or Pruijn et al (PMID: 7505620).

  • For the GEO dataset it is not specified how the data were processed (which background correction method, how normalized, etc…). Since this is not clear, the reliability of all conclusions based on this dataset can not be assessed.

  • Section 3.1

    • I am not convinced by the statements of this section. The expression values presented in table S1 show very little differences between the groups. E.g., Y1 N0+N1 = 7.69, N2+N3 = 7.83. Here the authors compare 126 versus 144 samples. In such a comparison even the faintest difference will be called “significant”, however a real biological influence can be doubted. Furthermore, the authors do not provide information on how the p-values were obtained. Did the authors perform a proper test for differential gene expression (connected to the above mentioned issue about data pre-processing)? How have the p-values been adjusted for multiple testing, if at all?

    • Figure 1 is a mess. The authors do not specifiy what groups of the respective staging levels they compare neither in the figure itself nor in the legend (they do it partly in the main text, however, this is not appropriate). One can only guess, that the colors might encode expression values, however there is no color legend given. In the legend the authors list a set of statistical tests they used, however, the reader is not informed which of them is used for which comparison. In Fig. 2 the authors present a very similar data representation, however, there they provide a color legend and in addition tables listing the data. Why this inconsistency?

  • Section 3.3

    • What is the rationale of shorten the observation time for survival analyses from 2500 to 1000 days? Just to get significant results?

  • Section 3.4

    • In the introductory sentence the authors write they used “ all genes derived from the examined GEO dataset” for correlation analyses. However, in the caption of Fig. 5 it is said that there correlation between yRNAs and “changed genes in HPV infection” are shown. What exactly is meant with “changed”?

    • The authors state that “a closer examination of protein coding genes correlated with YRNA1 showed that many of them are strictly correlated with HPV proteins” What does correlation with HPV proteins mean in detail? Is “many of them” statistically significant (more than expected by chance - hypergeometric test/Fisher’s exact test)?

  • Section3.5

    • I do not agree with the statement that the results from the GSEA “strongly” suggest a correlation with protein secretion. GSEA can give hints for hypotheses but should not be over-emphasized (especially with a NES of only -1.5)

  • Section 3.7

    • Regarding the overexpression of Y1: the used pcDNA vecor uses a CMV promoter recruting Pol II. Where in the MCS is the Y1 gene located or in other words, how many bases precede the actual start of the gene and are thus included in the transcribed RNA? Is there a plasmid map available?

    • Was DNAse digestion performed prior to the qRT-PCR? A Northern Blot would be more convincing.

    • “Analysis of RNA sequencing data of FaDu and Detroit562 cells overexpressing YRNA1 using the REACTOME pathway browser” – What was analyzed? Differentially expressed genes?

    • What is the point of looking at the most abundant genes from both cell lines in Fig. 9C

Minor concerns

  • In the discussion section the authors this time compared their results more extensively with other findings, with the most intensively considered study being a 2 year old paper from the same group showing mostly opposing results.

  • References 5 and 9 are identical

  • l145: “significantly altered genes from GSEA” – GSEA does not yield altered genes!

  • References for used tools like STAR and RSEM should be included

Author Response

Dear Reviewer,

Thank you very much for your kind and wise comments. Thanks to them our manuscript looks better than before. We have done our best to address each and every of your comment. We hope you will find our responses satisfactory and will accept our manuscript for publication. Thank you one more time. Please find the responses below.

Sincerely,

Kacper Guglas

In this re-submission from Guglas and colleagues, the authors present a revised study regarding noncoding yRNAs in HNSCC samples and cell lines and a putative connection to HPV infections. They analyzed total RNAseq data from cell lines overexpressing Y1 yRNA as well as a publicly available HNSCC microarray dataset and a polyA-enriched RNA seq HNSCC dataset from the TCGA project.

Major concerns:

  • Almost all analyses the authors mention are still described inappropriately. For some analyses it is not clear why the authors performed them at all.

RESPONSE:

We are not sure what is the issue since two other reviewers did not mention anything wrong in our analyses and they are pleased with them. We have changed the description in some sections and changed some figures to make them more transparent. We hope that it will be better now.

  • All results sections merely list some numbers without any conclusions or connections to the results of other subsections.

RESPONSE:

A small conclusion from each figure and analysis is always added at the end of the specific section, however more detailed conclusions are in discussion section. In results section we have focused strictly on the results and numbers.

  • The introduction and Discussion sections still contain inappropriately many self-citations. E.g., when the authors introduce yRNAs, they list 2 of their own review papers, however, they do not mention important original works, e.g. Hendrick et al (PMID: 6180298) or Pruijn et al (PMID: 7505620).
  • RESPONSE

Throughout the whole paper containing over 100 cited papers we have included only 5 of our previously published papers. We do not see this as an error, especially when this is the third paper in our Y RNA publication cycle. On previous review the reviewer asked for more diverse citations in the introduction and we have addressed this comment adding many, additional papers. We want to fulfill the reviewers suggestion so the two proposed papers are added to the main text and references.

  • For the GEO dataset it is not specified how the data were processed (which background correction method, how normalized, etc…). Since this is not clear, the reliability of all conclusions based on this dataset can not be assessed.

RESPONSE

We have provided the accession ID 65858 so that anyone interested further can check it for themselves. We did not copy all that information from already published paper in order not to generate repeats. We have added a sentence stating where exactly all that information may be found. We hope that our response will satisfy the reviewer.

  • Section 3.1

  • I am not convinced by the statements of this section. The expression values presented in table S1 show very little differences between the groups. E.g., Y1 N0+N1 = 7.69, N2+N3 = 7.83. Here the authors compare 126 versus 144 samples. In such a comparison even the faintest difference will be called “significant”, however a real biological influence can be doubted. Furthermore, the authors do not provide information on how the p-values were obtained. Did the authors perform a proper test for differential gene expression (connected to the above mentioned issue about data pre-processing)? How have the p-values been adjusted for multiple testing, if at all?

RESPONSE:

All statistical test were explained in Materials and Methods section and we have stated which test we have used in the figure legend.

  • Figure 1 is a mess. The authors do not specifiy what groups of the respective staging levels they compare neither in the figure itself nor in the legend (they do it partly in the main text, however, this is not appropriate). One can only guess, that the colors might encode expression values, however there is no color legend given. In the legend the authors list a set of statistical tests they used, however, the reader is not informed which of them is used for which comparison. In Fig. 2 the authors present a very similar data representation, however, there they provide a color legend and in addition tables listing the data. Why this inconsistency?

RESPONSE:

We have added the legend. Thank you for noticing that. We have tried also to make it more transparent. All the data considering the groups and values are available in the supplementary materials. However we have added an information in the figure legend that the values presented are from clustered analysis and we have stated in which cases exactly the analysis was clustered. All the significant values were described in the main text above the figure. When it comes to statistical tests we have listed all the statistical tests we have used. These differed in each analysis, it depended on how many variables were compared and on the outcome of the normality test. We have never before described it very detailed in the figure legend. The description of statistical test when is used which may be found in the Materials and Methods section.

  • Section 3.3

  • What is the rationale of shorten the observation time for survival analyses from 2500 to 1000 days? Just to get significant results?

RESPONSE:

Many researchers use 1000 days time in their OS and PFS analyses. That is why we also wanted to show these analyses in that “standard” time of 1000 days. We also wanted to show how the OS and PFS changes in terms of time taken under consideration for analysis.

  • Section 3.4
    • In the introductory sentence the authors write they used “ all genes derived from the examined GEO dataset” for correlation analyses. However, in the caption of Fig. 5 it is said that there correlation between yRNAs and “changed genes in HPV infection” are shown. What exactly is meant with “changed”?

RESPONSE:

The meaning behind this statement is that we have analysed all genes in the GEO data set and in the figure 5A we show the genes that are significantly correlated with each Y RNA. And these are the “changed” genes we have used for further analysis in this section. The caption of figure 5 was changed to avoid confusion.

  • The authors state that “a closer examination of protein coding genes correlated with YRNA1 showed that many of them are strictly correlated with HPV proteins” What does correlation with HPV proteins mean in detail? Is “many of them” statistically significant (more than expected by chance - hypergeometric test/Fisher’s exact test)?

RESPONSE:

In the next sentences it is described which HPV genes we meant. These are genes that code proteins essential for HPV. The correlation we meant is that genes that correlate with Y RNAs in terms of expression also can be found in many processes where HPV proteins are involved in or are found in the same processes that HPV proteins are. The phrase “many of them” is just a figure of speech, we did not calculate the statistics for that. We did not think these percents would be vital for this analysis.

  • 5

  • I do not agree with the statement that the results from the GSEA “strongly” suggest a correlation with protein secretion. GSEA can give hints for hypotheses but should not be over-emphasized (especially with a NES of only -1.5)

RESPONSE:

We have changed this statement to avoid confusion.

  • Section 3.7

  • Regarding the overexpression of Y1: the used pcDNA vecor uses a CMV promoter recruting Pol II. Where in the MCS is the Y1 gene located or in other words, how many bases precede the actual start of the gene and are thus included in the transcribed RNA? Is there a plasmid map available?

RESPONSE:

The exact plasmid is organized as this:

CMV promoter – 235-818

T7 promoter – 863-882

BamHI – 929-934

hRNY1 – 935-1047

The numbers above represent the position of each component in the plasmid. We also have the exact plasmid map if needed.

  • Was DNAse digestion performed prior to the qRT-PCR? A Northern Blot would be more convincing.

RESPONSE:

There was no need to do additional DNAse digestion, because the isolation kit uses columns which bind only RNA, resulting in extraction only the RNA, without the DNA fraction. We have performed an electrophoresis to confirm the absence of the DNA in our samples. Additionally if extracted RNA would not be pure, without DNA contamination, it would not have been used for NGS analysis. The same RNA was used for qRT-PCR. We do not have the possibility to perform a Northern Blot analysis at this moment.

  • “Analysis of RNA sequencing data of FaDu and Detroit562 cells overexpressing YRNA1 using the REACTOME pathway browser” – What was analyzed? Differentially expressed genes?

RESPONSE:

Significant genes derived from NGS analysis were further analysed in the REACTOME pathway browser to examine in which pathways these genes may be found.

  • What is the point of looking at the most abundant genes from both cell lines in Fig. 9C

RESPONSE:

We wanted to show which genes were the most abundant in this analysis, also to show differences/similarities between the cell lines and plasmids.

Minor concerns

  • In the discussion section the authors this time compared their results more extensively with other findings, with the most intensively considered study being a 2 year old paper from the same group showing mostly opposing results.

RESPONSE:

There is very limited number of previously published papers available in the field of Y RNAs and HNSCC and HPV infection. We have tried our best to find as many related papers as possible.

  • References 5 and 9 are identical

RESPONSE:

This mistake was corrected. Thank you for noticing that.

  • l145: “significantly altered genes from GSEA” – GSEA does not yield altered genes!

RESPONSE:

We have changed “significantly altered genes from GSEA” into “genes from GSEA”.

  • References for used tools like STAR and RSEM should be included

RESPONSE:

References used for STAR and RSEM tool were added.

Reviewer 2 Report (New Reviewer)

The manuscript made by Guglas K et al. It is interesting, well written, and well done.

The authors explain the association of YRNA 1,3,4,5 related with clinical parameters, life expectancy and their association with HPV and their status (active or inactive), despite of interesting manuscript I have a few suggestions that authors need to resolve.
Discussion
Line 552-553: These lines are interesting since the authors describe tumoral heterogeneity of HNSCC. Would it be possible describe intratumoral homogeneity related to HPV infection in HNSCC?

Lines 671-672: Would it be possible to discuss better their results related to molecular aggressiveness subtypes? (" Classical vs basal vs atypical vs mesenchymal") How is it the relationship between YRNA in each molecular aggressiveness subtype and progression?

Lines 743 - 744: These lines are interesting, since authors discuss the relationship between WNT b-catenin and TGF-b signaling in the HPV + group. Those pathways, mainly WNT b-catenin
 are related to EMT-phenomena, including HPV +. Would it be possible to do a hypothesis related to WNT b catenin, HPV +, EMT and YRNAs.

Lines 760-761: I suggest in these lines, describe the roles of YRNAs in the HNSCC development.

Overall comments
These manuscript is interesting, the methodology is extensive and well described, authors related the most important factors related to HNSCC aggressiveness and relationship with YRNAs, the methodology is completely related with results, authors provide enough tables, figures and supplementary material to understand the association between YRNA and HNSCC, these features impact directly to discussion that is well described. Authors describe the interaction of YRNAs in different malignant neoplasm and compare their results obtained. However, I suggest describing better the association between HNSCC and YRNAs due in some paragraphs, it seems to focus more on other tumors instead on HNSCC.

Author Response

Dear Reviewer,

Thank you very much for your kind and wise comments. Thanks to them our manuscript looks better than before. We have done our best to address each and every of your comment. We hope you will find our responses satisfactory and will accept our manuscript for publication. Thank you one more time. Please find the responses below.

Sincerely,

Kacper Guglas

The manuscript made by Guglas K et al. It is interesting, well written, and well done.

The authors explain the association of YRNA 1,3,4,5 related with clinical parameters, life expectancy and their association with HPV and their status (active or inactive), despite of interesting manuscript I have a few suggestions that authors need to resolve.
Discussion
Line 552-553: These lines are interesting since the authors describe tumoral heterogeneity of HNSCC. Would it be possible describe intratumoral homogeneity related to HPV infection in HNSCC?

RESPONSE:

We have added these information according to the reviewer’s suggestion.

Lines 671-672: Would it be possible to discuss better their results related to molecular aggressiveness subtypes? (" Classical vs basal vs atypical vs mesenchymal") How is it the relationship between YRNA in each molecular aggressiveness subtype and progression?

RESPONSE:

We have added these information according to the reviewer’s suggestion, however the knowledge is this field is limited making it difficult to discuss with other papers. As far as we know we are trying to discover something that nobody examined before and there is lack of similar articles.

Lines 743 - 744: These lines are interesting, since authors discuss the relationship between WNT b-catenin and TGF-b signaling in the HPV + group. Those pathways, mainly WNT b-catenin are related to EMT-phenomena, including HPV +. Would it be possible to do a hypothesis related to WNT b catenin, HPV +, EMT and YRNAs.

RESPONSE:

We have formed a hypothesis concerning YRNA5, EMT, WNT b catenin and HPV+. However this is just a hypothesis which need to be further analysed. We could not discuss it with any other previously published paper due to very limited knowledge in the field.

Lines 760-761: I suggest in these lines, describe the roles of YRNAs in the HNSCC development.

RESPONSE:

Every previously published information considering Y RNAs and HNSCC that we were able to find is already included in the manuscript. These studies are very novel considering the Y RNAs in HNSCC and Y RNAs and HPV infection.

Overall comments
These manuscript is interesting, the methodology is extensive and well described, authors related the most important factors related to HNSCC aggressiveness and relationship with YRNAs, the methodology is completely related with results, authors provide enough tables, figures and supplementary material to understand the association between YRNA and HNSCC, these features impact directly to discussion that is well described. Authors describe the interaction of YRNAs in different malignant neoplasm and compare their results obtained. However, I suggest describing better the association between HNSCC and YRNAs due in some paragraphs, it seems to focus more on other tumors instead on HNSCC.

RESPONSE:

As we have stated above, we have used every information considering HNSCC and Y RNAs that we were able to find. There are just no studies considering the topic and we are trying to describe something new for the science.

Reviewer 3 Report (New Reviewer)

The associations between Y RNAs and HPV infection in 2

HNSCC.= title need changes

Abstract: HPV infection is one of the most important risk factors for head and neck squamous cell 18

carcinoma among younger patients. Y RNAs are short non-coding RNAs involved in DNA replica- 19

tion. Y RNAs have been found to be dysregulated in many cancers, including head and neck squa- 20

mous cell carcinoma (HNSCC). In this study we investigated the role of Y RNAs in HPV positive 21

HNSCC using publicly available gene expression datasets from HNSCC tissue, where expression 22

patterns of Y RNAs in HPV(+) and HPV(-) HNSCC samples significantly differed. Y RNAs expres- 23

sion levels varied according to cancer pathological and clinical stages, and correlated with more 24

aggressive subtypes. Y RNAs were mostly associated with more advanced cancer stages in the 25

HPV(+) group, and YRNA3 and YRNA1 expression levels were found to be correlated with more 26

advanced clinical stages despite HPV infection status, showing that they may function as a potential 27

biomarker of more advanced stages of the disease. YRNA5 was associated with less advanced can- 28

cer stages in the HPV(-) group. Overall survival and progression-free survival analyses showed 29

opposite results between HPV groups. The expression of Y RNAs, especially YRNA1, correlated 30

with a vast number of proteins and cellular processes associated with viral infections and immuno- 31

logic response to viruses. HNSCC-derived cell lines overexpressing YRNA1 were then used to de- 32

termine the correlation of YRNA1 and the expression of genes associated with HPV infections. 33

Taken together results highlight the potential of Y RNAs as possible HNSCC biomarkers and new 34

molecular targets.

COMMENTS: Pls include more info on the Method used

Since Y RNAs are easily obtained from human serum, plasma, saliva and tissues, it 78

makes them potential biomarkers and targets for future therapies [13]. Studies have 79

shown that Y RNAs are over-expressed in glioma [24], triple negative breast cancer 80

(TNBC) [25], pancreatic ductal adenocarcinoma (PDAC) [26,27], colon cancer [28,29], cer- 81

vix cancer [26,28], benign prostate hyperplasia [30] and clear cell renal cell carcinoma 82

(ccRCC) [15]. On the other hand, Y RNAs have been found to be downregulated in 83

HNSCC [1], prostate cancer [30] and bladder cancer [31]. Y RNAs are involved in crucial 84

processes of cancer development such as apoptosis, cell proliferation, angiogenesis, me- 85

tastasis and different types of cellular stresses [13]. However, the involvement of Y RNAs 86

in viral-associated tumours, such as HPV infections in HNSCC, is unknown and their bi- 87

ological role is not defined yet. In order to address this role, Y RNAs (YRNA1, YRNA3, 88

YRNA4 and YRNA5) expression patterns were investigated in publicly available RNAseq 89

datasets of HPV(-) and HPV(+) HNSCC tissue samples. Additionally, HNSCC-derived 90

cell lines overexpressing YRNA1 were used to determine the correlation of YRNA1 and 91

the expression of genes associated with HPV infections. 92

COMMENT: Pls state the aim/objective of the study

2.1. HNSCC gene expression datasets

2.2. Y RNAs expression and clinical parameters

2.3. Functional enrichment analysis and prediction of gene function

2.4. Estimation of immune cells fractions

2.5. Statistical analysis

2.6. Cell line, transfection, RNA isolation and qRT-PCR

COMMENT: Pls restructure the methodology

3.1. The expression of YRNAs is significantly distinct between HNSCC clinical and pathological 234

Stages

3.2. Y RNAs are differently correlated with cancer and stemness markers

3.3. Patients' survival rate is associated with Y RNAs expression levels

3.4. Y RNAs are correlated with different genes among HPV(+) group with influence on HPV 408

proteins, viral and immunologic pathways

3.5. YRNA1 is strongly correlated with protein secretion processes

3.6. YRNA1 expression significantly correlates with immune cells

3.7. YRNA1 overexpression in HNSCC cell lines upregulate genes associated with responses to 488

viral infection

COMMENT: the subheading needs to be rephrased

In this study, YRNA1 was found to be correlated with HPV infection and immune 819

response to cancer disease. Results show a significant correlation of YRNA1 and HPV 820

proteins and immune processes. On the other hand, YRNA5 was found to be overex- 821

pressed only in the HPV(-) group, making it a potential biomarker on HPV infection status 822

in HNSCC. YRNA1 and YRNA3 were associated with more advanced cancer stages, and 823

YRNA5 was associated with less advanced cancer stages, suggesting a potential role of 824

these Y RNAs as biomarkers for HNSCC tumours Next, we found that the higher the 825

Biomedicines 2022, 10, x FOR PEER REVIEW27 of 31

expression of Y RNAs the more aggressive tumour subtype. Additionally, Y RNAs were 826

associated with cancer and stemness markers showing their negative correlation between 827

them, and opposite correlations between the most and the least aggressive subtypes. It 828

was also discovered that YRNA1 and YRNA4 may be potential prognostic biomarkers of 829

survival, differing between HPV(+) and HPV(-) groups of patients. Y RNAs also were 830

found to be enriched in a vast number of processes correlated with cancerogenesis, viral 831

and immunogenic pathways. The overexpression of YRNA1 in HNSCC-derived cell lines 832

confirmed the expression of genes co-expressed with YRNA1, and suggest a role for 833

YRNA1 in viral infections, including HPV infection in HNSCC patients. All these findings 834

show how Y RNAs may interfere in cancer progression, especially in association with 835

HPV infection, and should be evaluated as biomarkers and potential therapeutic targets.. 836

COMMENT: Pls summarise better and re-synthesize the conclusion

Author Response

Dear Reviewer,

Thank you very much for your kind and wise comments. Thanks to them our manuscript looks better than before. We have done our best to address each and every of your comment. We hope you will find our responses satisfactory and will accept our manuscript for publication. Thank you one more time. Please find the responses below.

Sincerely,

Kacper Guglas

The associations between Y RNAs and HPV infection in 2

HNSCC.= title need changes

 RESPONSE:

The title was changed to “The impact of YRNAs on HNSCC and HPV infection”.

Abstract: HPV infection is one of the most important risk factors for head and neck squamous cell 18

carcinoma among younger patients. Y RNAs are short non-coding RNAs involved in DNA replica- 19

tion. Y RNAs have been found to be dysregulated in many cancers, including head and neck squa- 20

mous cell carcinoma (HNSCC). In this study we investigated the role of Y RNAs in HPV positive 21

HNSCC using publicly available gene expression datasets from HNSCC tissue, where expression 22

patterns of Y RNAs in HPV(+) and HPV(-) HNSCC samples significantly differed. Y RNAs expres- 23

sion levels varied according to cancer pathological and clinical stages, and correlated with more 24

aggressive subtypes. Y RNAs were mostly associated with more advanced cancer stages in the 25

HPV(+) group, and YRNA3 and YRNA1 expression levels were found to be correlated with more 26

advanced clinical stages despite HPV infection status, showing that they may function as a potential 27

biomarker of more advanced stages of the disease. YRNA5 was associated with less advanced can- 28

cer stages in the HPV(-) group. Overall survival and progression-free survival analyses showed 29

opposite results between HPV groups. The expression of Y RNAs, especially YRNA1, correlated 30

with a vast number of proteins and cellular processes associated with viral infections and immuno- 31

logic response to viruses. HNSCC-derived cell lines overexpressing YRNA1 were then used to de- 32

termine the correlation of YRNA1 and the expression of genes associated with HPV infections. 33

Taken together results highlight the potential of Y RNAs as possible HNSCC biomarkers and new 34

molecular targets.

COMMENTS: Pls include more info on the Method used

RESPONSE:

More information considering the methods used was added to the abstract.

Since Y RNAs are easily obtained from human serum, plasma, saliva and tissues, it 78

makes them potential biomarkers and targets for future therapies [13]. Studies have 79

shown that Y RNAs are over-expressed in glioma [24], triple negative breast cancer 80

(TNBC) [25], pancreatic ductal adenocarcinoma (PDAC) [26,27], colon cancer [28,29], cer- 81

vix cancer [26,28], benign prostate hyperplasia [30] and clear cell renal cell carcinoma 82

(ccRCC) [15]. On the other hand, Y RNAs have been found to be downregulated in 83

HNSCC [1], prostate cancer [30] and bladder cancer [31]. Y RNAs are involved in crucial 84

processes of cancer development such as apoptosis, cell proliferation, angiogenesis, me- 85

tastasis and different types of cellular stresses [13]. However, the involvement of Y RNAs 86

in viral-associated tumours, such as HPV infections in HNSCC, is unknown and their bi- 87

ological role is not defined yet. In order to address this role, Y RNAs (YRNA1, YRNA3, 88

YRNA4 and YRNA5) expression patterns were investigated in publicly available RNAseq 89

datasets of HPV(-) and HPV(+) HNSCC tissue samples. Additionally, HNSCC-derived 90

cell lines overexpressing YRNA1 were used to determine the correlation of YRNA1 and 91

the expression of genes associated with HPV infections. 92

COMMENT: Pls state the aim/objective of the study

RESPONSE:

The aim/objective of the study was added to the Introduction section.

2.1. HNSCC gene expression datasets

2.2. Y RNAs expression and clinical parameters

2.3. Functional enrichment analysis and prediction of gene function

2.4. Estimation of immune cells fractions

2.5. Statistical analysis

2.6. Cell line, transfection, RNA isolation and qRT-PCR

COMMENT: Pls restructure the methodology

 RESPONSE:

The methodology section was previously restructurized with collaboration with dr Severino from Brasil. However we have slightly restructurized it again by placing the “Statistical Analysis” subsection at the end of the Methods section.

3.1. The expression of YRNAs is significantly distinct between HNSCC clinical and pathological 234

Stages

3.2. Y RNAs are differently correlated with cancer and stemness markers

3.3. Patients' survival rate is associated with Y RNAs expression levels

3.4. Y RNAs are correlated with different genes among HPV(+) group with influence on HPV 408

proteins, viral and immunologic pathways

3.5. YRNA1 is strongly correlated with protein secretion processes

3.6. YRNA1 expression significantly correlates with immune cells

3.7. YRNA1 overexpression in HNSCC cell lines upregulate genes associated with responses to 488

viral infection

COMMENT: the subheading needs to be rephrased

RESPONSE:

Some of the subheading were changed according to the reviewer’s suggestion.

In this study, YRNA1 was found to be correlated with HPV infection and immune 819

response to cancer disease. Results show a significant correlation of YRNA1 and HPV 820

proteins and immune processes. On the other hand, YRNA5 was found to be overex- 821

pressed only in the HPV(-) group, making it a potential biomarker on HPV infection status 822

in HNSCC. YRNA1 and YRNA3 were associated with more advanced cancer stages, and 823

YRNA5 was associated with less advanced cancer stages, suggesting a potential role of 824

these Y RNAs as biomarkers for HNSCC tumours Next, we found that the higher the 825

Biomedicines 2022, 10, x FOR PEER REVIEW27 of 31

expression of Y RNAs the more aggressive tumour subtype. Additionally, Y RNAs were 826

associated with cancer and stemness markers showing their negative correlation between 827

them, and opposite correlations between the most and the least aggressive subtypes. It 828

was also discovered that YRNA1 and YRNA4 may be potential prognostic biomarkers of 829

survival, differing between HPV(+) and HPV(-) groups of patients. Y RNAs also were 830

found to be enriched in a vast number of processes correlated with cancerogenesis, viral 831

and immunogenic pathways. The overexpression of YRNA1 in HNSCC-derived cell lines 832

confirmed the expression of genes co-expressed with YRNA1, and suggest a role for 833

YRNA1 in viral infections, including HPV infection in HNSCC patients. All these findings 834

show how Y RNAs may interfere in cancer progression, especially in association with 835

HPV infection, and should be evaluated as biomarkers and potential therapeutic targets.. 836

COMMENT: Pls summarise better and re-synthesize the conclusion

RESPONSE:

We have re-frased some parts of the conclusions.

This manuscript is a resubmission of an earlier submission. The following is a list of the peer review reports and author responses from that submission.

Round 1

Reviewer 1 Report

In this study author have discovered that YRNA1 is strongly correlated with HPV infection, HPV proteins, viral processes, and immune processes. Furthermore, YRNAs demonstrated their potential as possible HNSCC biomarkers and new molecular targets, especially YRNA1 showing properties of tumour suppressor. I think this manuscript has the potential to be published in this journal if the manuscript writing and presentation is thoroughly improved. For example, the background needs to have a better flow and the table legends should explain the table in detail. Proofreading the manuscript is a must to avoid errors in writing. The discussion section needs to be little modifications in a concise manner limited to the relevant findings of this study.

Reviewer 2 Report

In the current manuscript Guglas and colleagues present analyses regarding expression of noncodingyRNAs in HNSCC samples and cell lines and a putative connection to HPV infections. They analyzed total RNAseq data from cell lines overexpressing Y1 yRNA as well as a publicly available HNSCC microarray dataset and a polyA-enriched RNA seq HNSCC dataset from the TCGA project.  The authors conclude that Y1 is strongly correlated with HPV infection in HNSCC patient samples andthat Y1 plays a tumor suppressive role in this cancer entity and thus, Y1 may be a potential new biomarker or even therapeutic target for treating HNSCCs. Although the main hypotheses are quite interesting and the basic ideas sound quite convincing therealization and description of the analyses shows fundamental flaws. Transfection results are not shown at all, data analyses are not described properly and the figures are lacking clarity.

Major concerns:

• The introductory section is characterized by a high amount of self-citations regarding quite general topics like the investigated tumor entity HNSCC and non-coding RNAs. The quality of the Introduction would be improved substantially by incorporating references from original publications on these topics.

• There is no evidence provided that the transfection of Y1 was successful

• Data analyses in general were not described appropriately

â—¦ l. 135: “resulting in FASTQ files with reference to Human Genome hg19/GRC37, UCSC; annotations Gencode v29, Ensembl 90” – Fastq files contain sequence information along with bascalling quality mesures. They do not contain anything related to reference genomes/transcriptomes. This sentence raises several questions: How were sequencing reads aligned to the genome? How was summarization/quantification of mapped reads performed? What about Normalization? The mentioned reference genome is hg19/GRCh37, however, annotations (Gencode v29 and ENSEMBL 90 – why two,? When used which?) is based on GRCh38! Were there any downstream analyses (e.g., determination of differential gene experssion) and if so, how were they performed?

â—¦ Two different publicly available datasets of HNSCC patient sample were used, a microarray based set and polyA-enriched TCGA data. However, for me it was not clear, for which analysis which dataset was used. If the TCGA dataset was used for analyses regarding yRNA expression, the result are unreliably, since yRNA expression cannot be determined appropriately from a polyA-RNAseq.

â—¦ GSEA parameters are not explained, especially the question remains open what the authors used as input gene lists. L. 183: “significantly altered genes from GSEA” What does that mean? GSEA does not determine altered genes. In addition in l. 362ff the authors first mention enriched pathways but then they speak about enriched yRNAs. What do they really mean?

â—¦ “ The REACTOME database was used as a free tool for pathway analysis” A database is not a tool. Does this mean the authors used the over representation analysis provided via the REACTOME website?

â—¦ L. 226: it is not clear what the authors did to get the list of genes they investigated. Is it genes differentially expressed between untransfected and transfected cells? If yes, how did they determine differential gene expression (see above)? Or is it just the genes with the highest RNA abundance (however, it is also not stated how this was determined). If this is the case, then how can they conclude anything from this without comparing the expression levels with the untransfected cells?

â—¦ Figure 1: “most abundant genes from FaDu_hRNY1 and Detroit_hRNY1 cell lines were found using REACTOME database” What is the statement/sense behind this sentence?

â—¦ L. 536ff: “YRNA1 was found to be significantly upregulated in clustered analysis of

N2+N3 compared to N0+N1 (p = 0.0193).” What does clustered analysis mean? Only providing a p-value without the effect size tells nothing. Same for the nex sentence.

â—¦ In the materials and methods section, the authors mention that correlation analysis was performed using either Pearson or Spearman correlation coefficients. However, in the respective results parts it is not stated which method they used. Sometimes it can be derived from the figures, if there was a caption indicating Spearman correlation. In addition, correlation coefficients were always referred to as “r”, however, the correlation coefficient used in Spearman’s method is called “rho”/”ρ”.

â—¦ So, in total data analysis is very inadequately described and doesn’t allow a proper assessment of the presented results.

• Figures are not presented in a clear way. They should all be revised thoroughly to highlight the statements they were intended to show. Furthermore, the authors should consider to provide, at least part of some figures, at supplementary figures.

• One of the main conclusions of the manuscript is that Y1 is correlated with HPV infection in HNSCC patients, however, Figure 3 shows that there is no difference in any yRNA’s expression between HPV positive and negative patient samples. Although they later mentioned differences in expression between stages/lymph node metastasis status they do not provide a statement about this discrepancy with there main hypothesis.

The discussion is merely a summary of the results section without connecting the results into a broader contentand without discussing the many limitations of this work.

• There is no single validation of the presented results, so the whole manuscript is just a collection of hypotheses

Minor concerns

• Introduction, l. 65: although it is assumed that only ~2% of the human genome encodes for mRNAs, the remaining 98% is not entirely encoding for ncRNAs, there are also,e.g., regulatory sequences not transcribed at all.

• Introduction, l. 75 human YRNAs were already described in the 1980s, I would not consider this as a “novel described type of ncRNA”